# Comparative effects of two heat acclimation protocols consisting of high-intensity interval training in the heat on aerobic performance and thermoregulatory responses in exercising rats

Myla Aguiar Bittencourt[1,2], Samuel Penna Wanner 🔟[2]*, Ana Cançado Kunstetter[2], Nicolas Henrique Santos Barbosa[2], Paula Carolina Leite Walker[2], Pedro Victor Ribeiro Andrade[2], Tiago Turnes[1], Luiz Guilherme Antonacci Guglielmo[1]

**1** Physical Effort Laboratory, Graduate Program in Physical Education, Universidade Federal de Santa Catarina, Florianópolis, Santa Catarina, Brazil, **2** Exercise Physiology Laboratory, Graduate Program in Sport Sciences, Universidade Federal de Minas Gerais, Belo Horizonte, Minas Gerais, Brazil

* samuelwanner@ufmg.br

## Abstract

Acclimation resulting from low- to moderate-intensity physical exertion in the heat induces several thermoregulatory adaptations, including slower exercise-induced increases in core body temperature. However, few studies have investigated the thermoregulatory adaptations induced by high-intensity interval training (HIIT) protocols. Thus, the present study aimed to compare the adaptations in rats' thermoregulatory parameters and aerobic performance observed after two different heat acclimation regimens consisting of HIIT protocols performed in a hot environment. Twenty-three adult male Wistar rats were initially subjected to an incremental-speed exercise at 32°C until they were fatigued and then randomly assigned to one of the following three heat acclimation strategies: passive heat exposure without any exercise (untrained controls–UN; n = 7), HIIT performed at the maximal aerobic speed ($HIIT_{100\%}$; n = 8) and HIIT performed at a high but submaximal speed ($HIIT_{85\%}$; n = 8). Following the two weeks of interventions, the rats were again subjected to a fatiguing incremental exercise at 32°C, while their colonic temperature ($T_{COL}$) was recorded. The workload performed by the rats and their thermoregulatory efficiency were calculated. After the intervention period, rats subjected to both HIIT protocols attained greater workloads ($HIIT_{100\%}$: 313.7 ± 21.9 J vs. $HIIT_{85\%}$: 318.1 ± 32.6 J vs. UN: 250.8 ± 32.4 J; p < 0.01) and presented a lower ratio between the change in $T_{COL}$ and the distance travelled ($HIIT_{100\%}$: 4.95 ± 0.42°C/km vs. $HIIT_{85\%}$: 4.33 ± 0.59°C/km vs. UN: 6.14 ± 1.03°C/km; p < 0.001) when compared to UN rats. The latter finding indicates better thermoregulatory efficiency in trained animals. No differences were observed between rats subjected to the two HIIT regimens. In conclusion, the two HIIT protocols induce greater thermoregulatory adaptations and performance improvements than passive heat exposure. These adaptations do not differ between the two training protocols investigated in the present study.

**Data Availability Statement:** All relevant data are within the manuscript and its Supporting Information files.

**Funding:** The authors are indebted to the Conselho Nacional de Desenvolvimento Científico e Tecnológico (CNPq; grant number 312374/2018-6; www.cnpq.br/). This study was financed in part by the Coordenação de Aperfeiçoamento de Pessoal de Nível Superior - Brasil (CAPES) - Finance Code 001. MAB and ACK were recipients of PhD fellowships from the CAPES (https://www.capes.gov.br/), whereas NHSB and PVRA were recipients of undergraduate fellowships from the Fundação de Amparo à Pesquisa do Estado de Minas Gerais (FAPEMIG; https://fapemig.br/). SPW received funding from Pró-Reitoria de Pesquisa, Universidade Federal de Minas Gerais (BR) (PRPq - UFMG 2020). The funders had no role in study design, data collection and analysis, decision to publish, or preparation of the manuscript.

**Competing interests:** We have read the journal's policy and the authors of this manuscript have the following competing interests: Samuel Penna Wanner, who is the corresponding author of this manuscript, currently works as an Academic Editor for PLOS One. The other authors have declared that no competing interests exist.

## Introduction

Acclimation is defined as physiological (autonomic) or behavioural adaptations occurring within an organism caused by experimentally induced stressful changes in particular climatic factors [1]. These adaptations allow organisms to reduce strain or enhance tolerance when exposed to extreme environments. In this context, heat acclimation results from repeated exposure to artificial conditions that induce whole-body hyperthermia and profuse sweating (in the case of humans), thereby promoting adaptations aimed at improving tolerance to heat stress and reducing thermoregulatory strain and the risk of developing heat-related disorders [2,3].

Considering the above-mentioned benefits, heat acclimation protocols are often employed by athletes to improve their performance during competitions under conditions of environmental heat stress. The best acclimation strategy to increase human performance in the heat is the association between physical exercise and exposure to hot ambient temperature [2,4]. A seminal study by Cohen & Gisolfi [4] showed that aerobic training in temperate environments slightly reduced physiological strain and improved tolerance during exercise-heat stress; however, these improvements were modest compared to the large improvements that followed heat acclimation resulting from 8 exercise sessions in the heat.

Heat acclimation induces several thermoregulatory adaptations, including a lower resting core body temperature and an increased cutaneous heat loss, as evidenced by a greater sweat rate and skin blood flow, as well as lower core temperature thresholds for activating these thermoeffectors [2,5]. These adaptations result in slower and attenuated exercise-induced increases in core temperature and heart rate [4,6]. Collectively, the above-mentioned adaptations and the resulting lower physiological strain markedly improve aerobic performance in hot environments [7,8].

Traditional heat acclimation protocols are characterised by low-to moderate-intensity physical exertion, with individuals exercising at intensities close to 45–60% of their maximal oxygen uptake ($\dot{V}O_{2max}$) [8–10]. Nevertheless, recent studies have reported performance improvements and greater thermal comfort while exercising in the heat after different heat acclimation protocols consisting of high-intensity interval training (HIIT) performed at 80–100% of $\dot{V}O_{2max}$ [11–14]. Notably, these HIIT protocols represent a more time-efficient strategy than protocols based on low- or moderate-intensity exercises and may better reflect the specific requirements of most sport activities [11,13].

The studies regarding heat acclimation induced by HIIT protocols did not investigate thoroughly the ensuing thermoregulatory adaptations [11–14]. Considering that, for a given duration, a bout of high-intensity exercise produces greater increases in core temperature than low-intensity exercise both in humans [15,16] and rats [17,18], it is expected that the higher thermoregulatory strain during HIIT sessions might be more effective to induce heat acclimation. However, the relationship among changes in core temperature, exercise intensity and exercise duration may not be so simple under uncompensable heat stress conditions observed when the rate of metabolic heat production exceeds the body's physiological ability to dissipate heat, thus leading to a continuous increase in core temperature [2]. Under these conditions, tolerance to high-intensity exercise is compromised and, therefore, lower intensities, which are tolerated for longer periods [19,20], may even enhance the increase in core temperature [20] and induce greater thermoregulatory strain.

Here, we investigated short-term adaptations induced by two HIIT protocols with different number, intensity and duration of training sessions. It is relevant to say that the development of well-controlled protocols using laboratory animals is an important step to study

thermoregulatory adaptations induced by HIIT, particularly those involving the central nervous system and/or those that require the use of invasive techniques to be revealed.

Thus, the present study aimed to compare the adaptations in rats' thermoregulatory parameters and aerobic performance observed during and after two different heat acclimation protocols consisting of HIIT regimens performed in a hot environment. Our first hypothesis was that the two HIIT protocols investigated would induce greater thermoregulatory adaptations and performance improvements than passive heat exposure (control group). Our second hypothesis concerned comparisons between HIIT protocols. We expected that a HIIT protocol characterised by higher running speeds would induce more noticeable improvements in performance because predominant overload on the intensity of training sessions has resulted in greater tolerance to aerobic exercise than predominant overload on the duration of training sessions [21]. In contrast, we expected that the greater workload associated with a HIIT protocol characterised by lower running speeds would induce augmented hyperthermia during training sessions and, therefore, would induce more evident thermoregulatory adaptations.

## Materials and methods

### Ethical approval

All experiments were approved by the local Ethics Commission on Animal Use of the Universidade Federal de Minas Gerais (protocol 364/2016) and were conducted in accordance with the regulations provided by the Brazilian National Council for the Control of Animal Experimentation. Efforts were made to minimise the number of rats used and their suffering.

### Animals

Twenty-three male Wistar rats ($287 \pm 7$ g at the start of experiments; approximately 2 months old) were purchased from the Institute of Biological Sciences at the Universidade Federal de Minas Gerais. These rats were housed collectively in groups of four in polypropylene cages at a room temperature of 24˚C, under 14/10 h light/dark cycles (lights on at 5:00 am) and had free access to water and rat chow.

### Experimental design

The rats were initially subjected to a 5-day familiarisation protocol with running on a treadmill. Next, the rats were subjected to a first incremental-speed exercise (incremental exercise, for simplicity's sake) and were randomly assigned into one of the following three groups consisting of different heat acclimation strategies: passive heat exposure without any exercise (untrained control rats–UN; n = 7), high-intensity interval training performed at the maximal aerobic speed ($HIIT_{100\%}$; n = 8), high-intensity interval training performed at a high but submaximal speed ($HIIT_{85\%}$; n = 8). Forty-eight hours after the first incremental exercise, the heat acclimation strategies were initiated. They lasted 2 weeks and, 48 h after the last session, the groups performed again an incremental exercise to evaluate the effects of heat acclimation on thermoregulatory responses and aerobic performance. During all the acclimation sessions and incremental exercises, the rats had their colonic temperature ($T_{COL}$) and tail-skin temperature ($T_{TAIL}$) measured, and the ambient temperature was controlled at 32˚C (hot environment). This ambient temperature was selected because rats running at 32˚C exhibited lower aerobic performance and greater exercise-induced increases in brain and abdominal temperatures compared with rats running at 25˚C [22]. Following the second incremental test, the rats were euthanised with an intraperitoneal injection of a lethal dose of anaesthetic (ketamine 240 mg/kg and xylazine 31.5 mg/kg).

## Procedures

**Familiarisation with running on a treadmill.** The rats were familiarised with running on a custom-made treadmill designed for small animals (Gaustec Magnetismo; Brazil) during 5 consecutive days (Table 1). This protocol was used to teach the rats the direction to run. During the last three familiarisation sessions, the rats ran with a thermocouple attached to their tail and a thermistor inserted into their colon; this procedure allowed us to train the rats to run while simultaneously preventing them from becoming entangled in the thermocouple and thermistor wires [23]. All rats used in this study exhibited a steady running pattern with minimal exposure to electrical stimuli (0.5 mA) during the last familiarisation session.

**Incremental exercise.** The rats were taken from their home cages to the experimental room, where a thermocouple was taped to their tail and a thermistor was inserted into their colon. Immediately after these procedures, the rats were placed on the treadmill, and the exercise was started. As recently reported, the act of subjecting rats to treadmill running without a previous resting period is an adequate and time-saving method for measuring exercise-induced thermoregulatory responses in this species [24].

The incremental exercises were started at a speed of 10 m/min, and this speed was increased by 1 m/min every 3 min until the rats were fatigued [25]. Fatigue was defined as the moment when the animals were no longer able to keep pace with the treadmill and thus were exposed to electrical stimulation for 10 s [26]. Physical performance measured during this exercise (i.e., workload) has been shown to be positively correlated with the $\dot{V}O_{2MAX}$ of untrained rats [21]. The maximal aerobic speed ($S_{MAX}$) attained during the first incremental exercise was used to prescribe the intensity of training sessions in the two HIIT regimens.

**Passive heat exposure.** The rats in the UN group were exposed to the hot environment, without being exercised, for the same time as the rats subjected to the $HIIT_{85\%}$ protocol (the one with the more prolonged sessions between the two training regimens studied).

**HIIT protocols.** Two different HIIT regimens were performed in the heat. The $HIIT_{100\%}$ was adapted from the protocol proposed by Rahimi et al. [27] and was characterised by 2-min running at 95–100% of $S_{MAX}$ separated by 2-min intervals consisting of active recovery at 65% of $S_{MAX}$ (Table 2). During the first session, the rats completed four exercise-recovery cycles. One cycle was added every session (except the fifth session when intensity was increased from 95% to 100% of $S_{MAX}$), so that the rats completed nine cycles during the last (i.e., seventh) session.

In contrast, $HIIT_{85\%}$ was adapted from the protocol proposed by Rolim et al. [28] and was characterised by 4-min running at 85% of $S_{MAX}$ separated by 2-min intervals consisting of active recovery at 65% of $S_{MAX}$ (Table 3). These rats completed four exercise-recovery cycles during the first session and one cycle was added every two sessions. At the last (i.e., tenth) session, these rats completed eight cycles. The two HIIT regimens lasted 2 weeks; there was a 48-h recovery between training sessions during $HIIT_{100\%}$, while a typical 24-h recovery was allowed during $HIIT_{85\%}$ (except the 72-h interval between the fifth and sixth sessions).

**Table 1. Five-day familiarisation protocol with running on a treadmill.**

| 1st day | 2nd day | 3rd, 4th and 5th days |
|---|---|---|
| 5 min (rest) 1 min (10 m/ min) 1 min (12 m/ min) 5 min (15 m/ min) | 5 min (rest) 1 min (12 m/ min) 5 min (15 m/ min) | insertion of a thermistor into the colon + attachment of a thermocouple to the tail surface 5 min (rest) 5 min (15 m/min) |

**Table 2. Description of training sessions of the HIIT_{100%}.**

| Day | Physical exertion | | | Recovery | | |
| --- | --- | --- | --- | --- | --- | --- |
| | Intensity (% $S_{MAX}$) | Duration (min) | Cycles | Intensity (% $S_{MAX}$) | Duration (min) | Total duration (min) |
| 1—Monday | 95 | 2 | 4 | 65 | 2 | 14 |
| 2—Wednesday | 95 | 2 | 5 | 65 | 2 | 18 |
| 3—Friday | 95 | 2 | 6 | 65 | 2 | 22 |
| 4—Sunday | 95 | 2 | 7 | 65 | 2 | 26 |
| 5—Tuesday | 100 | 2 | 7 | 65 | 2 | 26 |
| 6—Thursday | 100 | 2 | 8 | 65 | 2 | 30 |
| 7—Saturday | 100 | 2 | 9 | 65 | 2 | 34 |

HIIT_{100%} = high-intensity interval training at 100% of maximal speed; $S_{MAX}$ = maximal aerobic speed. The treadmill incline was kept constant at 5% during all training sessions and incremental exercises. Each training session was started with a 5-min warm-up run at 65% of $S_{MAX}$ (not computed in the total duration of the session).

## Measured and calculated variables

**Performance variables.** The $S_{MAX}$ attained and the workload performed by the rats during the incremental exercises (i.e., before and after interventions) were calculated. $S_{MAX}$ was calculated according to the following equation: $S_{MAX} = S + (t_1 / t_2)$, where S = speed in the last completed stage in m/min; $t_1$ = time spent in the incomplete stage in seconds; and $t_2$ = duration of each stage, which corresponds to 180 seconds [17].

The workload, which takes the body mass of rats into account, was considered as the index of aerobic performance and calculated as follows:

$$workload \ (J) = m.g.s.sin\theta.t$$

where m = body mass in kg; g = force of gravity (9.8 m/s$^2$); s = speed in m/min; $\theta$ = angle of treadmill inclination (5°); and t = time spent in each stage [21]. The workload values were calculated for each stage of the incremental exercise, including the incomplete stage, and were then summed; the value obtained after the summation corresponded to the exercise workload.

**Thermoregulatory variables.** During all the incremental exercises and acclimation sessions, $T_{COL}$ and $T_{TAIL}$ were registered every minute. $T_{TAIL}$ was measured as an indirect index

**Table 3. Description of training sessions of the HIIT_{85%}.**

| Day | Physical exertion | | | Recovery | | |
| --- | --- | --- | --- | --- | --- | --- |
| | Intensity (% $S_{MAX}$) | Duration (min) | Cycles | Intensity (% $S_{MAX}$) | Duration (min) | Total duration (min) |
| 1—Monday | 85 | 4 | 4 | 65 | 2 | 22 |
| 2—Tuesday | 85 | 4 | 4 | 65 | 2 | 22 |
| 3 –Wednesday | 85 | 4 | 5 | 65 | 2 | 28 |
| 4—Thursday | 85 | 4 | 5 | 65 | 2 | 28 |
| 5—Friday | 85 | 4 | 6 | 65 | 2 | 34 |
| 6—Monday | 85 | 4 | 6 | 65 | 2 | 34 |
| 7—Tuesday | 85 | 4 | 7 | 65 | 2 | 40 |
| 8—Wednesday | 85 | 4 | 7 | 65 | 2 | 40 |
| 9—Thursday | 85 | 4 | 8 | 65 | 2 | 46 |
| 10—Friday | 85 | 4 | 8 | 65 | 2 | 46 |

HIIT_{85%} = high-intensity interval training at 85% of the maximal speed; $S_{MAX}$ = maximal aerobic speed. The treadmill incline was kept constant at 5% during all training sessions and incremental exercises. Each training session was started with a 5-min warm-up run at 65% of $S_{MAX}$ (not computed in the total duration of the session).

of cutaneous blood flow, using a thermocouple (YSI 400 series reusable probe, Yellow Springs Instruments, USA) taped to the lateral surface, 1 cm from the base of the tail. Recently, a strong linear correlation was demonstrated between skin blood flow and surface temperature in the rat's tail; thus, the magnitude of tail artery vasodilation was reflected by corresponding increases in $T_{TAIL}$ [29]. $T_{COL}$ was measured using a lubricated thermistor probe (model 4491RJ, Measurement Specialties, USA) inserted 7 cm past the anal sphincter, after faecal pellets had been removed from the colon by gentle, external massage. $T_{COL}$ was taken as an index of core body temperature. As evidenced by a previous study, both telemetry and the rectal probe methods yielded similar responses to drugs that modulate thermoregulation and there were high positive correlations between the measurements provided by the two methods. [30]. Both the $T_{COL}$ and $T_{TAIL}$ probes were connected to a thermometer (YSI 4600 precision thermometer, Yellow Springs Instruments, USA), where the temperature values were displayed.

The increase in $T_{COL}$ was calculated by subtracting final from initial $T_{COL}$ during the incremental exercises. Next, the ratio between the increase in $T_{COL}$ and the distance travelled by the rats (˚C/km) was calculated. This ratio is inversely related to thermoregulatory efficiency; therefore, the lower the ratio, the higher the efficiency [31].

The ambient temperature inside the treadmill chamber was measured every minute using two thermocouples attached, with impermeable adhesive tape, to the ceiling of the acrylic chamber containing the treadmill belt; one thermocouple was placed at the front end and the other at the rear end of the chamber. The control of ambient temperature at 32˚C was attained with the help of two electric heaters (Britânia, PR, Brazil) that were positioned at the same level, one in front of and the other behind the treadmill belt.

**Body mass.** Body mass was measured three times a week (i.e., on Mondays, Wednesdays and Fridays), always between 4:00 and 5:00 pm, as an index of hydration status and overall animal health during the heat acclimation protocols. Body mass was also measured before the incremental exercises.

### Statistical analysis

Shapiro-Wilk's and Levene's tests were used to assess the normality and homoscedasticity of the data, respectively. Because all data presented a normal distribution, they were expressed as means ± standard errors of the mean (SEM). The curves showing the responses of $T_{COL}$ and $T_{TAIL}$ across exercise time points in rats of different groups were analysed using two-way analyses of variance (ANOVAs), with repeated measures applied only for time points. The average ambient temperature, body mass, workload, initial $T_{COL}$, final $T_{COL}$, change in $T_{COL}$ and final $T_{TAIL}$ were compared between groups (i.e., $HIIT_{100\%}$, $HIIT_{85\%}$ and UN groups) and moments (before and after the 2-week period) using two-way ANOVAs, with repeated measures applied only for moments. Body mass gain and the changes in workload or thermoregulatory efficiency induced by interventions were compared between groups using one-way ANOVAs. Whenever applicable, Tukey's *post hoc* tests were used to identify differences between pairs of means. The total workload performed by trained rats was compared between groups ($HIIT_{100\%}$ vs. $HIIT_{85\%}$) using an unpaired Student's *t* test. The significance level was set at $\alpha < 0.05$.

Effect size analyses were performed to compare the magnitude of differences in workload and thermoregulatory efficiency between the two groups subjected to HIIT protocols. We have calculated the Cohen's d effect-size (*ES*) by subtracting the mean value of one group from the mean value of the group it was being compared to. The result was then divided by a combined standard deviation for the data. The *ES* values were classified as trivial (*ES* < 0.2), small

(*ES* = 0.2–0.6), moderate (*ES* = 0.6–1.2), large (*ES* = 1.2–2.0), very large (*ES* = 2.0–4.0) or extremely large (*ES* ≥ 4.0) [32].

## Results

### Ambient temperature and body mass

The average ambient temperature did not differ between groups and incremental exercises or heat acclimation sessions (p > 0.05 for all comparisons; Table 4), indicating that we could successfully control ambient temperature within the desired range. Before the interventions, the $HIIT_{85\%}$ group had a lower body mass than the UN group, and this difference persisted after two weeks ($HIIT_{85\%}$: from 263.5 ± 7.1 g to 297.1 ± 8.9 g vs. UN: from 304.7 ± 16.4 g to 340.4 ± 11.6 g; $F_{GROUP}$ = 3.80, p < 0.05). However, the body mass gain during the intervention period was not different between the three groups ($HIIT_{100\%}$: 39.5 ± 7.4 g vs. $HIIT_{85\%}$: 33.6 ± 3.8 g vs. UN: 35.7 ± 16.1 g; $F_{GROUP}$ = 0.09, p > 0.05).

### Thermoregulatory parameters during the heat acclimation sessions

The workload during the training sessions in the $HIIT_{100\%}$ and $HIIT_{85\%}$ groups increased across the two weeks ($F_{MOMENT}$ = 1051.7; p < 0.001, Fig 1A). The $HIIT_{85\%}$ group performed greater work than $HIIT_{100\%}$ at the three moments evaluated: the first, middle and last training sessions ($F_{GROUP}$ = 7.43; p < 0.05). Moreover, the total workload performed by the rats during the two weeks was 67% greater in the $HIIT_{85\%}$ group than in the $HIIT_{100\%}$ group (p < 0.001; *ES* = 4.63; Fig 1B); this difference in total workload between the two trained groups can be classified as an extremely large effect size.

The $T_{COL}$ and $T_{TAIL}$ exhibited marked increases during all heat acclimation sessions in the three groups. The UN rats presented similar $T_{COL}$ kinetics during the first and the last heat exposure sessions; this temperature increased in the early moments of exposure and then reached a steady-state value between 38°C and 39°C. $T_{COL}$ analysis indicated significant group × moment interactions in the first (Fig 2A; $F_{INTER}$ = 15.18; p < 0.001) and last (Fig 2B; $F_{INTER}$ = 24.30; p < 0.001) acclimation sessions. The rats subjected to both HIIT protocols had higher $T_{COL}$ values when compared to the UN rats. The latter statement is valid for the first (Fig 2A) and last (Fig 2B) training sessions. No differences in $T_{COL}$ between $HIIT_{85\%}$ vs. $HIIT_{100\%}$ were observed whatsoever during these sessions (p > 0.05). With regard to $T_{TAIL}$, there were also significant group × moment interactions in the first (Fig 2C; $F_{INTER}$ = 3.89; p < 0.001) and last (Fig 2D; $F_{INTER}$ = 2.36; p < 0.01) acclimation sessions. Despite these significant interactions, the *post hoc* analyses only detected transient differences between groups (e.g., a higher $T_{TAIL}$ in $HIIT_{85\%}$ than in UN and $HIIT_{100\%}$) during the last session (p < 0.001).

Initial $T_{COL}$ did not differ between groups ($F_{GROUP}$ = 1.59; p > 0.05; Fig 3A), but decreased from the first to the last session ($F_{MOMENT}$ = 8.46, p < 0.05; Fig 3A); no group × moment interaction was observed for initial $T_{COL}$ ($F_{INTER}$ = 0.37; p > 0.05). In contrast, a significant group × moment interaction was observed for final $T_{COL}$ ($F_{INTER}$ = 9.68, p < 0.001), so that the $HIIT_{100\%}$ and $HIIT_{85\%}$ groups had higher temperatures than the UN group at the end of the two moments evaluated (e.g., last session: $HIIT_{100\%}$: 41.03 ± 0.22°C vs. $HIIT_{85\%}$: 40.90 ± 0.21°C vs. UN: 38.43 ± 0.22°C; p < 0.05). When comparing the sessions within a group, both the $HIIT_{100\%}$ and $HIIT_{85\%}$ rats presented higher final $T_{COL}$ in the last than in the first session, whereas the UN rats presented lower values in the last session. Similarly, a significant interaction was observed for the change in $T_{COL}$ ($F_{INTER}$ = 6.45, p < 0.01), which was higher in the two HIIT groups than in the UN group. Again, only the rats subjected to HIIT had a higher exercise-induced increase in $T_{COL}$ in the last as compared to the first session (Fig 3C).

**Table 4. Average ambient temperature (˚C) during the incremental exercises and during the first, middle and last training sessions.**

| Groups | Pre-Incremental exercise | Post- Incremental exercise | First session | Middle session | Last session |
|---|---|---|---|---|---|
| HIIT$_{100\%}$ | 32.10 ± 0.04 | 32.10 ± 0.05 | 32.10 ± 0.15 | 32.17 ± 0.03 | 32.07 ± 0.05 |
| HIIT$_{85\%}$ | 32.09 ± 0.04 | 32.24 ± 0.05 | 32.35 ± 0.34 | 32.13 ± 0.09 | 32.19 ± 0.14 |
| UN | 32.08 ± 0.06 | 32.09 ± 0.06 | 32.20 ± 0.10 | 31.99 ± 0.13 | 31.95 ± 0.21 |

HIIT$_{100\%}$: high-intensity interval training at 100% of maximal speed; HIIT$_{85\%}$: high-intensity interval training at 85% of maximal speed; UN: untrained, passive heat exposure. Data are expressed as the means ± SEM. The average ambient temperature was calculated from data collected every minute during exercise or passive exposure. The middle session corresponds to the fourth session in the HIIT$_{100\%}$ group and fifth session in the HIIT$_{85\%}$ and UN groups.

With regard to cutaneous heat loss, the final $T_{TAIL}$ was significantly influenced by the moment ($F_{MOMENT}$ = 24.56, p < 0.001) and group ($F_{GROUP}$ = 3.68, p < 0.05) (Fig 3D), although no interaction was observed between these factors ($F_{INTER}$ = 0.05, p > 0.05). Despite the observed main effect of group, the *post hoc* test could not identify differences during pairwise comparisons. Therefore, final $T_{TAIL}$ tended to be higher in the HIIT groups than in the UN group in the two moments evaluated (e.g., last session: HIIT$_{100\%}$: 36.97 ± 0.28˚C vs. HIIT$_{85\%}$: 36.78 ± 0.29˚C vs. UN: 35.57 ± 0.44˚C; Fig 3D; 0.05 < p < 0.10). In addition, final $T_{TAIL}$ was higher in the last than in the first session in all groups.

To better understand the thermoregulatory changes during the training sessions, we calculated the ratio between the change in $T_{COL}$ and the distance travelled by rats (Fig 4). Of note, this ratio is inversely correlated with thermoregulatory efficiency. Significant moment and group effects were observed for the $T_{COL}$-to-distance ratio (Fig 4A; $F_{GROUP}$ = 7.21, p < 0.05; $F_{MOMENT}$ = 108.91, p < 0.001), although no significant interaction was observed for these two factors ($F_{INTER}$ = 2.25, p > 0.05). The ratio reduced in the two HIIT groups during the last training session (p < 0.001), and the HIIT$_{100\%}$ group presented higher values than the HIIT$_{85\%}$ group in all training sessions (p < 0.05). For example, the ratio was 33% higher in HIIT$_{100\%}$ than in HIIT$_{85\%}$ during the last session (*ES* = 1.26); this difference can be classified as a large effect size.

When the ratio was analysed over all training sessions, the HIIT$_{100\%}$ and HIIT$_{85\%}$ groups presented reduced values relative to the 1st session in the 2nd ($F_{MOMENT}$ = 23.99; p < 0.001; Fig 4B) and 5th ($F_{MOMENT}$ = 12.18; p < 0.001; Fig 4C) sessions, respectively. Because the number of sessions evaluated by ANOVAs was different in the two trained groups (10 sessions in HIIT$_{85\%}$ and 7 in HIIT$_{100\%}$), we decided to perform an ANOVA using only the first 7 sessions of the HIIT$_{85\%}$ group. This additional analysis revealed that a reduced ratio between the change in $T_{COL}$ and the distance travelled was observed again in the 5th compared to the 1st session; therefore, the temporal difference regarding the changes in thermoregulatory efficiency between groups (as reported above) was not a mathematical artefact.

## Physical performance and thermoregulatory parameters during incremental exercises performed before and after the heat acclimation sessions

The statistical analysis revealed a significant group × moment interaction for workload ($F_{INTER}$ = 8.39; p < 0.01; Fig 5A). The workload performed during the first incremental exercise (pre-interventions) was not different between groups, and only the HIIT groups performed greater work in the second than in the first incremental exercise. Supporting the previous data, the changes induced by interventions were similar in the HIIT groups, but greater in these groups than in the UN group (HIIT$_{100\%}$: 86.5 ± 21.5 J vs. HIIT$_{85\%}$: 130.4 ± 20.4 J vs. UN: 7.9 ± 20.9 J; $F_{GROUP}$ = 8.39; p < 0.01; Fig 5B). However, despite the lack of statistical

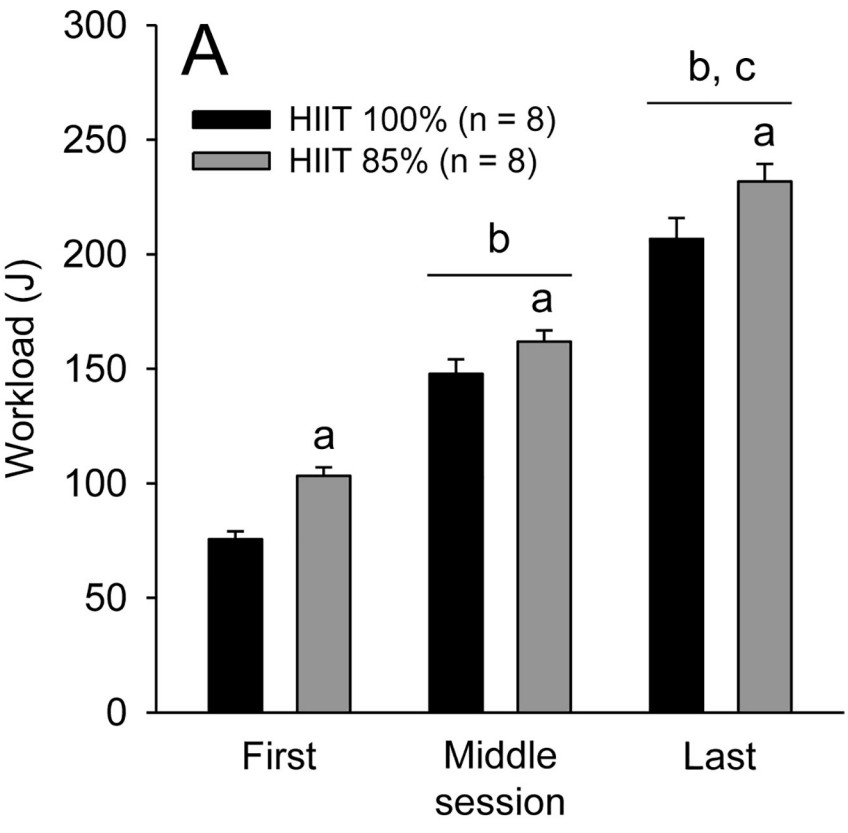

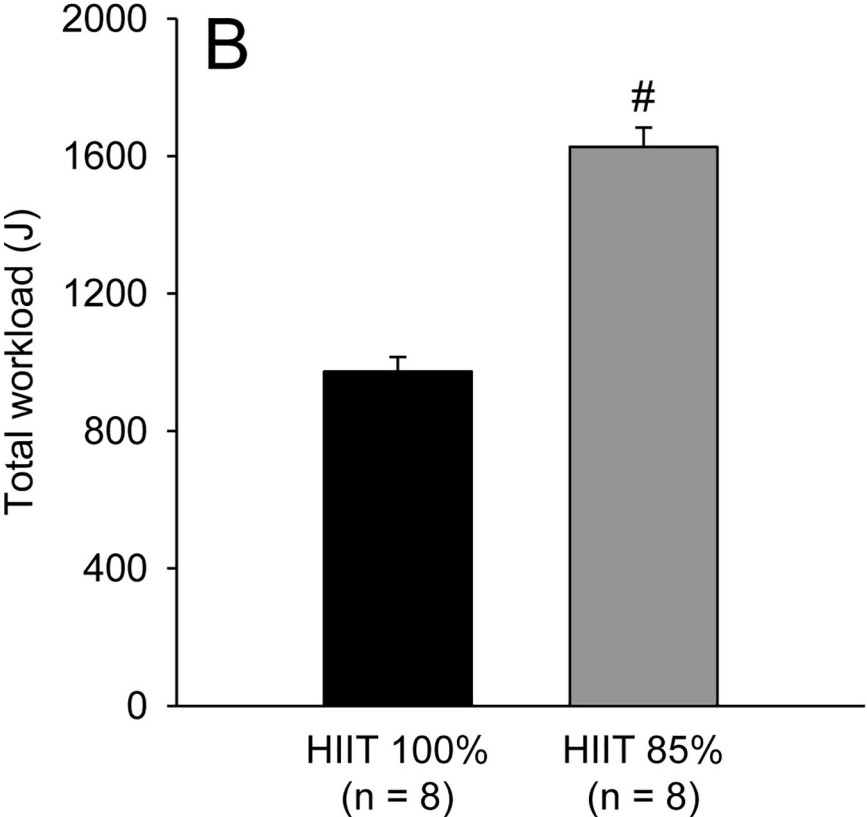

**Fig 1.** Workload performed by the rats subjected to the two HIIT protocols during the first, middle and last training sessions (panel A). Total workload performed during all training sessions by rats of the two trained groups (panel B). The values are expressed as the means ± SEM. $HIIT_{100\%}$: high-intensity interval training at 100% of maximal speed; $HIIT_{85\%}$: high-intensity interval training at 85% of maximal speed. The letter a denotes a main effect of group (i.e., significantly different from $HIIT_{100\%}$, irrespective of the moment; $p < 0.05$); the letters b and c denote a main effect of moment (i.e., significantly different from the first and middle sessions, irrespective of the group; $p < 0.001$); # denotes a significant difference from the $HIIT_{100\%}$ group ($p < 0.001$).

differences between groups subjected to HIIT, the change induced by intervention was 51% greater in $HIIT_{85\%}$ than in $HIIT_{100\%}$ ($ES = 0.74$); this greater value in the $HIIT_{85\%}$ group can be classified as a moderate effect size.

As expected, $T_{COL}$ and $T_{TAIL}$ were markedly increased during the first incremental exercise (at the pre-intervention moment; $p < 0.001$; Fig 6A and 6C). It is worth noting that the kinetics of these increases were similar in the three experimental groups. However, during the second incremental exercise (at the post-intervention moment), there was a group × time point interaction for $T_{COL}$ ($F_{INTER} = 2.73$; $p < 0.001$; Fig 6B), although the *post hoc* analysis could not reveal significant differences between pairs of means. Similar to the observations made in the first incremental exercise, the $T_{TAIL}$ during the second exercise was influenced by the time ($F_{TIME} = 163.40$; $p < 0.001$; Fig 6D), but not by the group factor ($F_{GROUP} = 0.97$; $p > 0.05$). Thus, ten minutes after the exercise had begun, the rats of the three groups exhibited progressive increases in $T_{TAIL}$, which persisted elevated until they were fatigued.

We also compared initial $T_{COL}$, $T_{COL}$ at fatigue, changes in $T_{COL}$ and $T_{TAIL}$ at fatigue between groups and moments (i.e., first vs. second incremental exercise). No significant main effect of group and no significant group × moment interactions were observed in any of the four variables analysed. Significant main effect of moment was observed for initial $T_{COL}$ ($F_{MOMENT} = 10.42$; $p < 0.01$; Fig 7A) and changes in $T_{COL}$ ($F = 5.56$; $p < 0.05$; Fig 7C), but not for $T_{COL}$ at fatigue ($F = 0.13$; $p > 0.05$; Fig 7B) or $T_{TAIL}$ at fatigue ($F = 0.94$; $p > 0.05$; Fig 7D).

We also calculated the ratio between the change in $T_{COL}$ and the distance travelled for the data collected during the incremental exercises before and after the 2-week interventions. A significant group × moment interaction was revealed ($F_{INTER} = 11.39$; $p < 0.001$), with the *post-hoc* analysis indicating a significant difference from pre- to post-intervention only in the $HIIT_{85\%}$ group ($p < 0.05$). Moreover, there were no differences between $HIIT_{100\%}$ vs. $HIIT_{85\%}$ and between $HIIT_{85\%}$ vs. UN (Fig 8A; $p > 0.05$). Because this analysis could not reveal clear differences between groups, we calculated the pre-to-post-intervention changes in the ratio. This way, both HIIT groups exhibited negative changes in the ratio that were different from the positive change reported for the UN group ($F_{GROUP} = 11.39$; $p < 0.001$). However, despite the lack of statistical differences between groups subjected to HIIT, the change induced by intervention was 165% greater in $HIIT_{85\%}$ than in $HIIT_{100\%}$ ($ES = 1.39$); this greater value in the $HIIT_{85\%}$ group can be classified as a large effect size.

## Discussion

The current study investigated the effects of passive exposure to a hot environment and of two different HIIT protocols performed in the same hot environment on thermoregulatory responses and aerobic performance in rats. As expected, the rats subjected to the HIIT protocols exhibited greater workload and thermoregulatory efficiency in the post-intervention incremental exercise than the rats passively exposed to heat, thus confirming our first hypothesis. We also hypothesized that a HIIT characterised by higher running speeds would induce greater performance improvements but less thermoregulatory adaptations; this second hypothesis was partially confirmed. As indicated by traditional statistics (i.e., ANOVAs), the

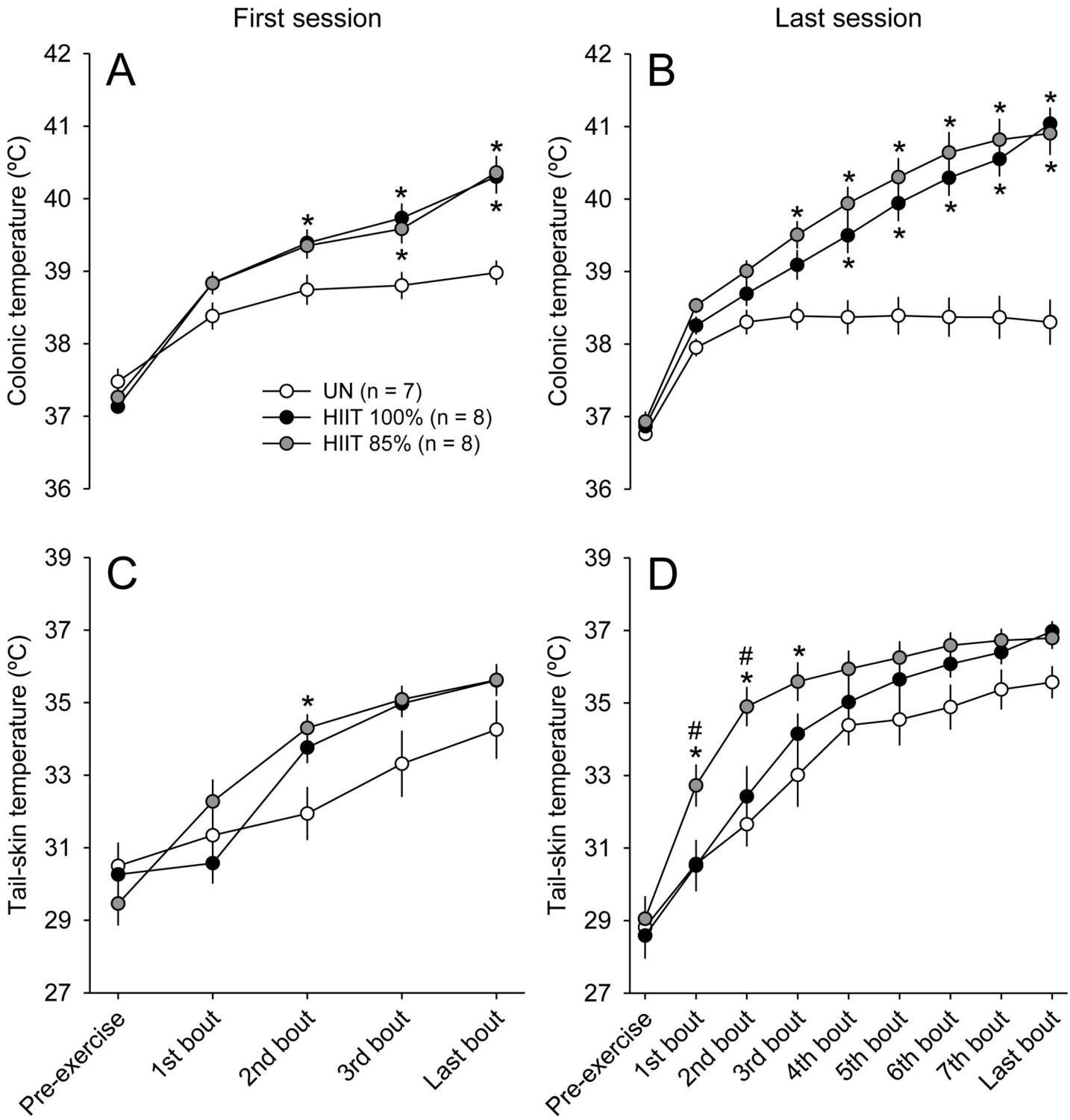

**Fig 2.** Colonic temperature ($T_{COL}$; panels A and B) and tail-skin temperature ($T_{TAIL}$; panels C and D) in rats of the three experimental groups during the first (A and C) and last (B and D) heat acclimation sessions. The values are expressed as the means ± SEM. $HIIT_{100\%}$: high-intensity interval training at 100% of maximal speed; $HIIT_{85\%}$: high-intensity interval training at 85% of maximal speed; UN: untrained, passive heat exposure. * denotes a significant difference from the UN group in the same bout ($p < 0.05$); # denotes a significant difference from the $HIIT_{100\%}$ group in the same bout ($p < 0.05$).

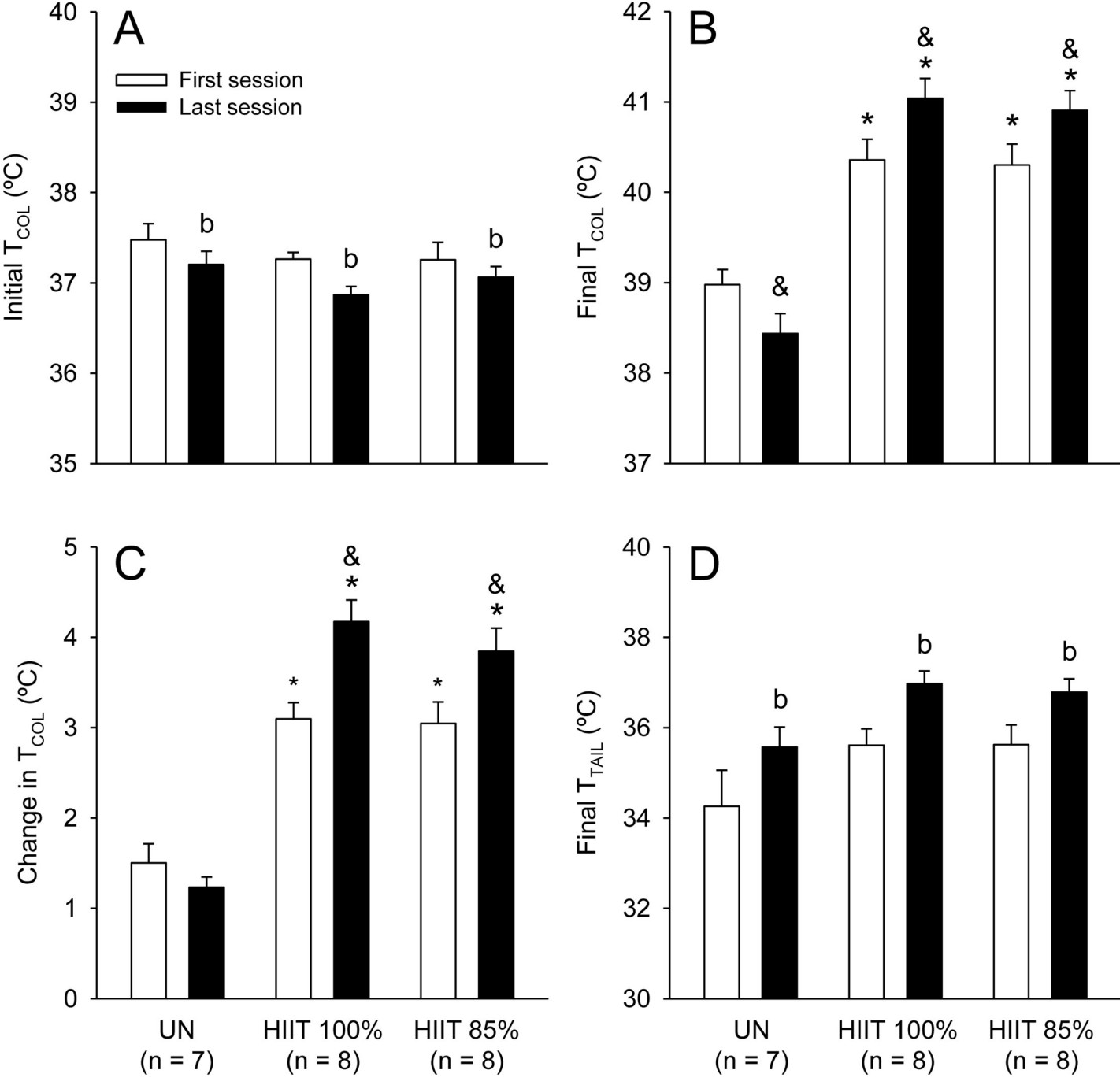

**Fig 3.** Initial (panel A) and final (panel B) colonic temperature ($T_{COL}$), change in $T_{COL}$ (panel C) and final tail-skin ($T_{TAIL}$) temperature (panel D) in rats of the three experimental groups in the first and last acclimation sessions. The values are expressed as the means ± SEM. $HIIT_{100\%}$: high-intensity interval training at 100% of maximal speed; $HIIT_{85\%}$: high-intensity interval training at 85% of maximal speed; UN: untrained, passive heat exposure. The letter b denotes a main effect of moment (i.e., significantly different from the first session, irrespective of the group; $p < 0.05$); * denotes a significant difference from the UN group in the same acclimation session ($p < 0.05$); & denotes a significant difference from the first acclimation session within the same group ($p < 0.05$).

gains in performance and thermoregulatory efficiency were similar in the two trained groups. In contrast, effect size analysis revealed more evident improvements in rats subjected to $HIIT_{85\%}$ than those subjected to $HIIT_{100\%}$ (moderate and large effect sizes, respectively).

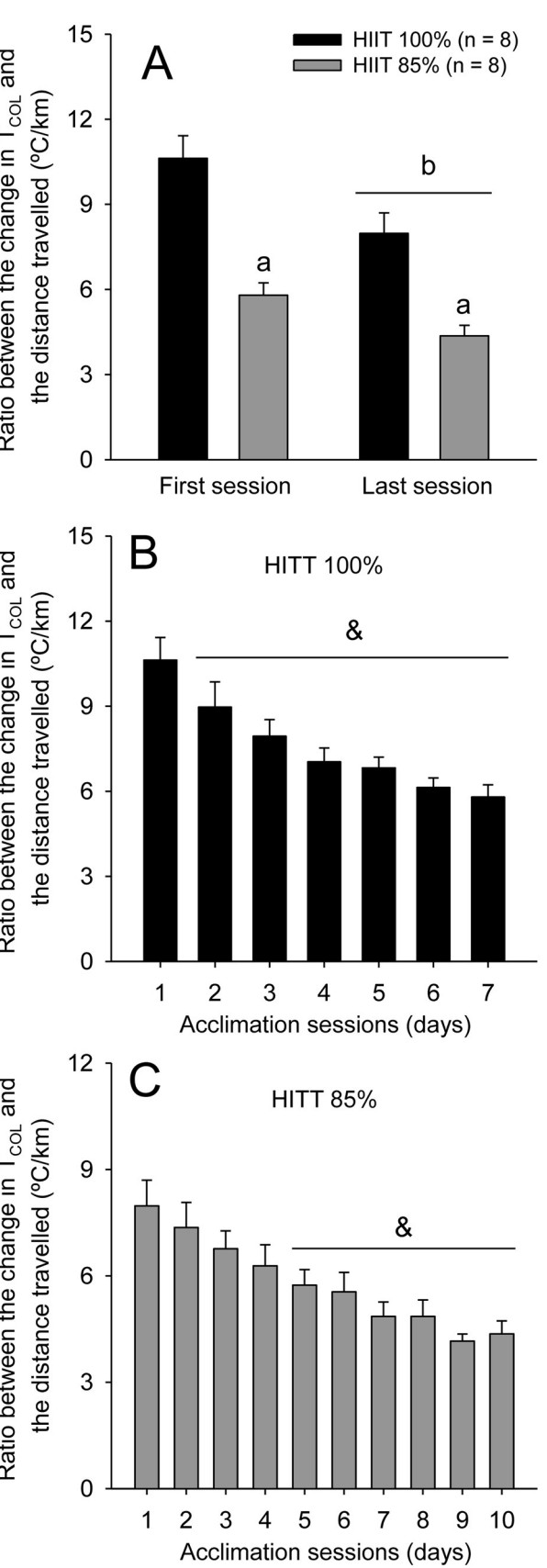

**Fig 4.** Ratio between the change in colonic temperature ($T_{COL}$) and the distance travelled in rats of the two HIIT groups during the first and last training sessions (panel A). Ratio between the change in $T_{COL}$ and the distance travelled across all training sessions in rats subjected to the $HIIT_{100\%}$ (panel B) or $HIIT_{85\%}$ (panel C) protocols. The values are expressed as the means ± SEM. $HIIT_{100\%}$: high-intensity interval training at 100% of maximal speed; $HIIT_{85\%}$: high-intensity interval training at 85% of maximal speed; The letter a denotes a main effect of group (i.e., significantly different from $HIIT_{100\%}$, irrespective of the moment; $p < 0.05$); the letter b denotes a main effect of moment (i.e., significantly different from the first session, irrespective of the group; $p < 0.05$); & denotes a significant difference from the first incremental exercise within the same group ($p < 0.05$).

The UN rats did not show a performance improvement or clear thermoregulatory adaptations following 2 weeks of passive heat exposure. The only adaptation observed was the decreased initial $T_{COL}$; however, this result allows different interpretations, as discussed later. Therefore, it is reasonable to suggest that 10 heat exposure sessions lasting between 27 and 51 min did not represent a thermal stimulus that was sufficiently strong enough to acclimate rats. A previous investigation, in which mice were exposed for a more prolonged period to a higher ambient temperature (i.e., 5 days of continuous passive heat exposure to 37˚C), could successfully identify adaptations that demonstrated the occurrence of heat acclimation [33]. The different species used and the differences in exposure duration and ambient temperature help to explain why our study diverges from the previously published study. It is worth noting that the effects of continuous heat exposure to 37˚C on aerobic performance in mice were not studied [33].

In order to identify the most effective heat acclimation regimen, this study compared the effects of two different HIIT protocols performed in a hot environment. Because the rats' body mass increased during the intervention period and was different between groups, we calculated the workload to analyse the changes in aerobic performance. Our results showed that both HIIT protocols increased the workload performed by the rats during the post-intervention incremental exercise. When the training-induced increase in the workload was compared between the two trained groups, no difference was observed (as evidenced by ANOVA), being the increase possibly more evident in $HIIT_{85\%}$ than in $HIIT_{100\%}$ (as evidenced by a moderate effect size), thus contradicting our second hypothesis. Few studies have addressed whether the volume and intensity of interval training sessions influence aerobic performance improvements in rats. A recent study [21] reported that, during 8 weeks of treadmill running at constant speeds in a temperate environment, predominant overloads in training intensity provided better performance improvements than predominant overloads in training duration (even though the distance travelled per session was always the same in the groups mentioned earlier). In the present study, the running intensities and durations and distance travelled differed between the two groups and, therefore, it is likely that the expected superior performance benefits resulting from higher intensities in the $HIIT_{100\%}$ group were limited by shorter running durations.

We investigated the effect of different exercise-heat strain caused by different training regimens on thermoregulatory adaptations. The $HIIT_{100\%}$ sessions were associated with lower thermoregulatory efficiency, despite no changes in $T_{COL}$ and minor changes in $T_{TAIL}$ (observed in the last training session), compared to the $HIIT_{85\%}$ sessions. When thermoregulation was compared between trained groups in the second incremental exercise, the initial $T_{COL}$ was reduced in all of them, with no intergroup differences in $T_{COL}$ at fatigue and in $T_{TAIL}$ whatsoever during the exercise. In addition, we did not observe a difference between groups when the training-induced improvement in thermoregulatory efficiency was compared using an ANOVA. Interestingly, the improvement was possibly more evident in $HIIT_{85\%}$ than in $HIIT_{100\%}$, as evidenced by a large effect size. This effect size analysis partially confirms our second hypothesis and highlights that the intergroup difference in thermoregulatory efficiency

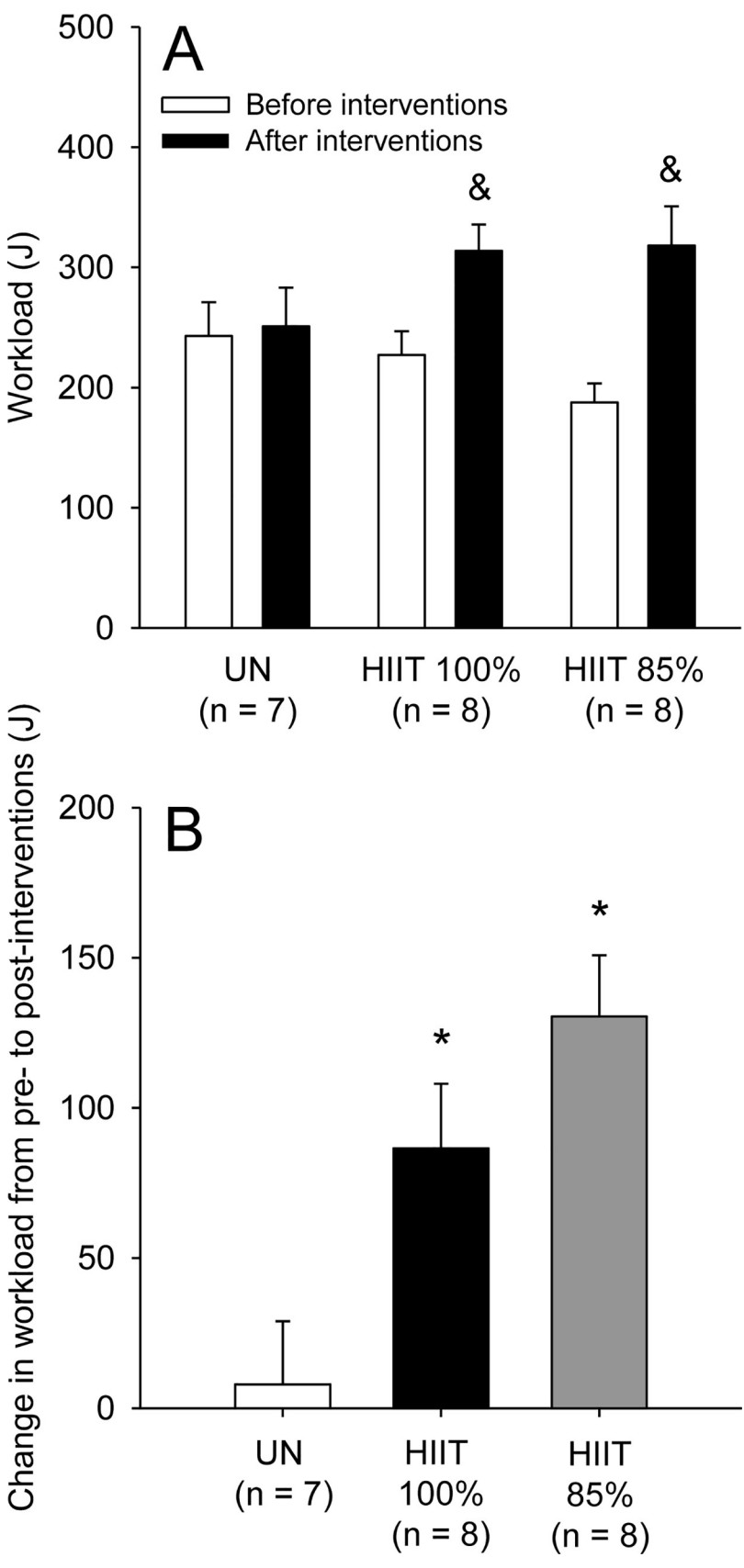

**Fig 5.** Workload performed by the rats of the three experimental groups during the incremental exercises at pre- and post-interventions (panel A). Change in workload from pre- to post-interventions (panel B). The values are expressed as the means ± SEM. HIIT$_{100\%}$: high-intensity interval training at 100% of maximal speed; HIIT$_{85\%}$: high-intensity interval training at 85% of maximal speed; UN: untrained, passive heat exposure. & denotes a significant difference from the first incremental exercise within the same group (p < 0.05); * denotes a significant difference from the UN group (p < 0.05).

during training sessions did not influence the thermoregulatory adaptations observed after the interventions.

Although thermoregulatory efficiency improved with fewer sessions in HIIT$_{100\%}$ rats, a better efficiency was not observed after the 2-week interventions. These results suggest that the superior exercise intensities performed by HIIT$_{100\%}$ rats during the training sessions may have been counterbalanced (maybe overcome) by the 60% higher workload performed by HIIT$_{85\%}$ rats during the 2 weeks. Another important point was the fact that HIIT$_{85\%}$ rats were subjected to longer heat exposures during training sessions; in this sense, exposure duration seems to be one of the stimuli underlying the development of heat acclimation. To isolate and identify the contribution of exercise intensity or duration, future experiments should compare the adaptations induced by HIIT protocols consisting of training sessions at different intensities but with the same total workload.

The observation that two HIIT protocols performed at 32°C induced comparative heat acclimation suggests that the endothermic heat production during exercise and the resulting body heat storage may drive acclimation in rats. In this sense, thermoregulatory adaptations will likely result from different combinations among environmental heat stress, exercise intensity and exercise duration that cause marked hyperthermia for long periods of time. Thus, although the comparison of effectiveness between HIIT and moderate-intensity continuous training was not an objective of the present study, our findings indicate that different training regimens, depending on how training sessions are planned, might be effective in inducing heat acclimation in rats. This idea is supported by an earlier study in trained subjects, in which 7 exercise sessions at 50% $\dot{V}O_{2MAX}$ for 60 min/day or at 75% $\dot{V}O_{2MAX}$ for 30–35 min/day (both at an ambient temperature of 40°C) similarly reduced heart rate and rectal temperature during a heat tolerance test [34].

In all training sessions or incremental exercises, the environmental heat stress combined with running intensities and duration resulted in uncompensable heat stress, a condition characterised by an inability to dissipate the metabolic heat produced, leading to marked increases in core temperature [2,35]. Regardless of the HIIT protocol, a plateau in T$_{COL}$ was never seen in the above-mentioned conditions. Thus, the adaptations induced by heat acclimation were not sufficiently strong to reverse these conditions of uncompensable heat stress, even though we cannot rule out that trained rats may have become more tolerant to the adverse outcomes caused by severe hyperthermia.

The initial T$_{COL}$ was reduced in the trained groups during the 2-week intervention. However, the same effect was observed in UN rats, suggesting that all three protocols, including passive heat exposure, have induced this adaptation, which was also reported in heat-acclimated humans [8–10]. Lower resting abdominal temperatures have also been observed in rats subjected to an 8-week aerobic training regime consisting of constant-speed sessions performed at 23°C [36]. However, the existence of lower resting T$_{COL}$ in heat-acclimated rats should be analysed with caution, considering that rats were quickly manipulated to insert the colonic thermistor and attach the tail-skin thermocouple before placement on the treadmill. As handling and placement on the treadmill represent psychological stressors that increase core temperature in this species [24], we cannot rule out that reduced initial T$_{COL}$ resulted

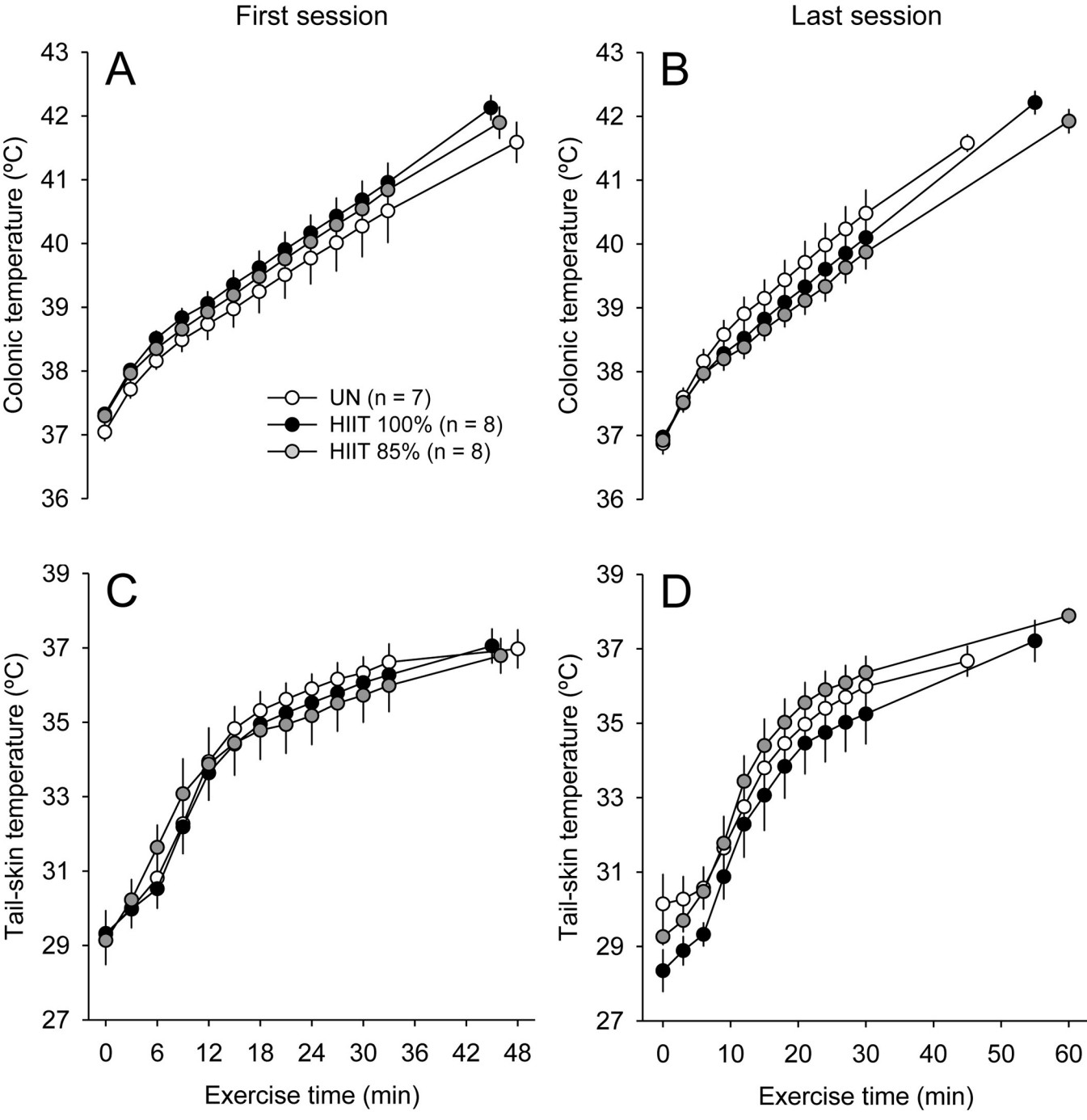

**Fig 6.** Colonic temperature ($T_{COL}$; panels A and B) and tail-skin temperature ($T_{TAIL}$; panels C and D) in rats of the three experimental groups during the incremental exercises at pre- (A and C) and post-interventions (B and D). The values are expressed as the means ± SEM. $HIIT_{100\%}$: high-intensity interval training at 100% of maximal speed; $HIIT_{85\%}$: high-intensity interval training at 85% of maximal speed; UN: untrained, passive heat exposure.

from habituation to these stressors. Whether the reduction in resting $T_{COL}$ is a heat acclimation-mediated adaptation or habituation to handling stress/exposure to an unpleasant environment should be further investigated by measuring circadian fluctuation of core temperature in acclimated rats using telemetry.

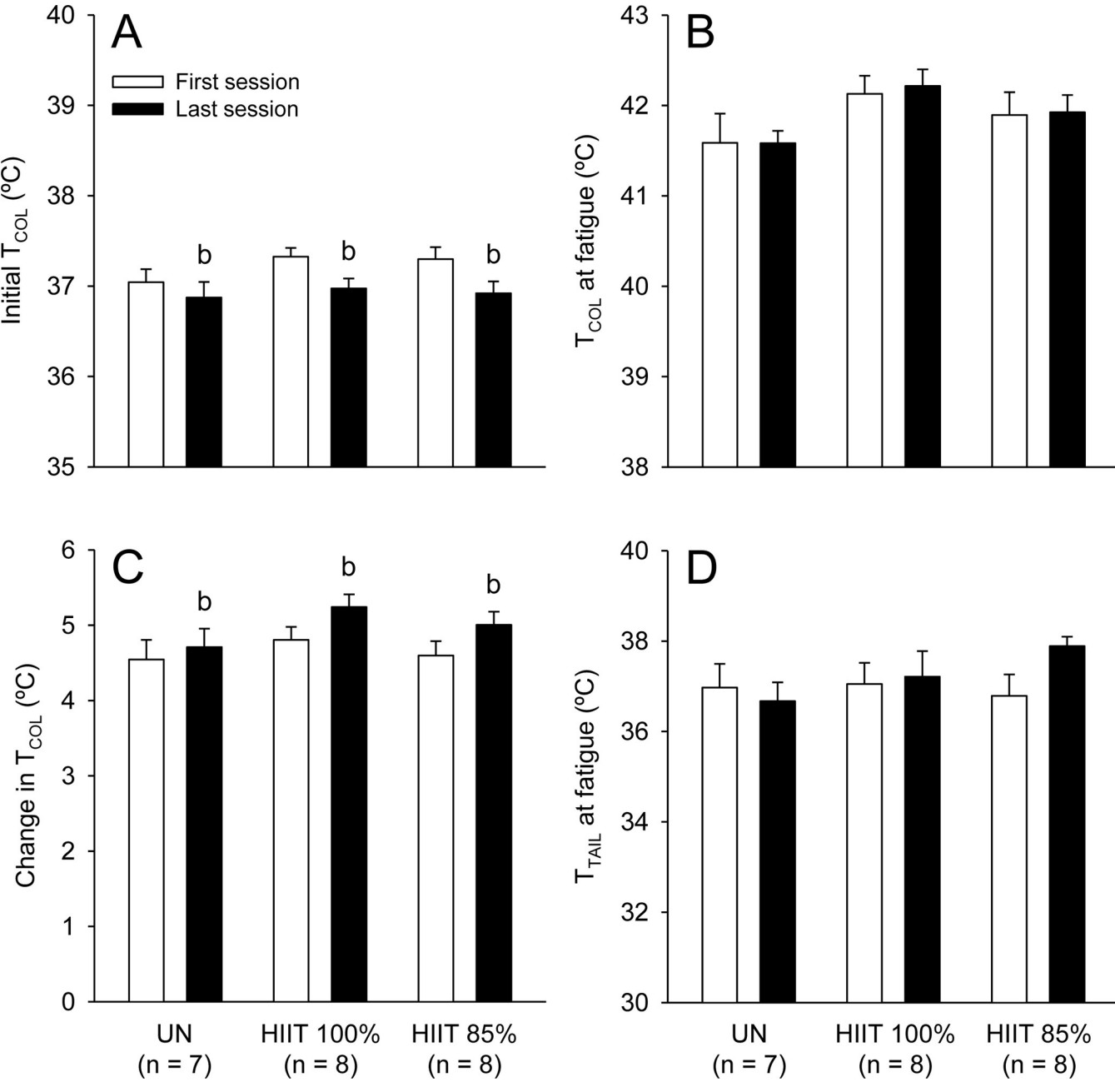

**Fig 7.** Initial colonic temperature ($T_{COL}$; panel A), $T_{COL}$ at fatigue (panel B), change in $T_{COL}$ (panel C) and tail-skin ($T_{TAIL}$) temperature at fatigue (panel D) in rats of the three experimental groups in the incremental exercises at pre- and post-interventions. The values are expressed as the means ± SEM. HIIT$_{100\%}$: high-intensity interval training at 100% of maximal speed; HIIT$_{85\%}$: high-intensity interval training at 85% of maximal speed; UN: untrained, passive heat exposure. The letter b denotes a main effect of moment (i.e., significantly different from the first session, irrespective of the group; $p < 0.05$).

The $T_{COL}$ measured at fatigue during the post-intervention incremental exercise was not different in the three groups. However, the time to reach these temperature levels was longer in the two HIIT groups compared with the UN group. This means that the HIIT protocols increased thermoregulatory efficiency in rats, because they have presented the same exercise-induced increase in $T_{COL}$ despite performing greater work. Previous studies have reported that heat-acclimated subjects exercise for longer periods before attaining a core temperature of

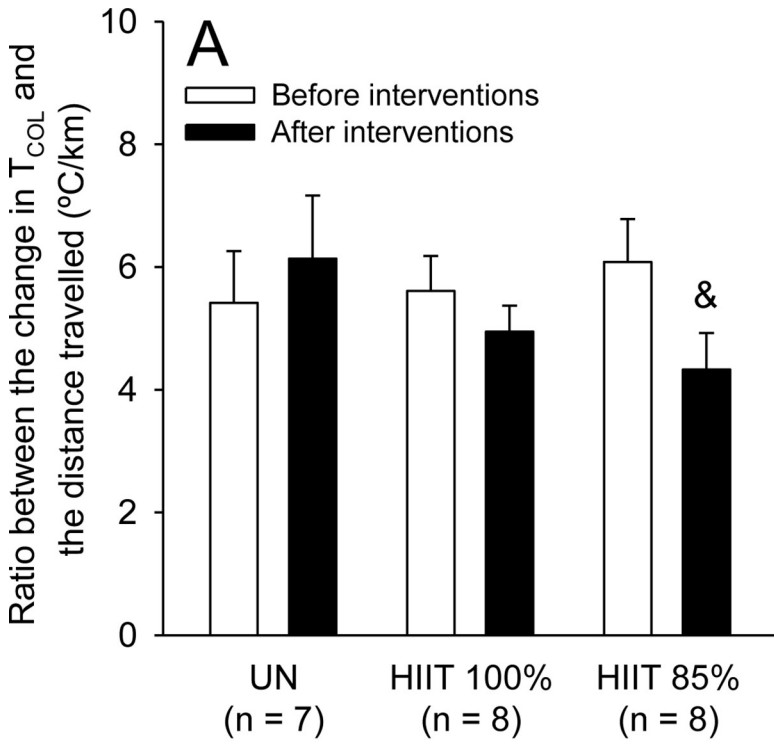

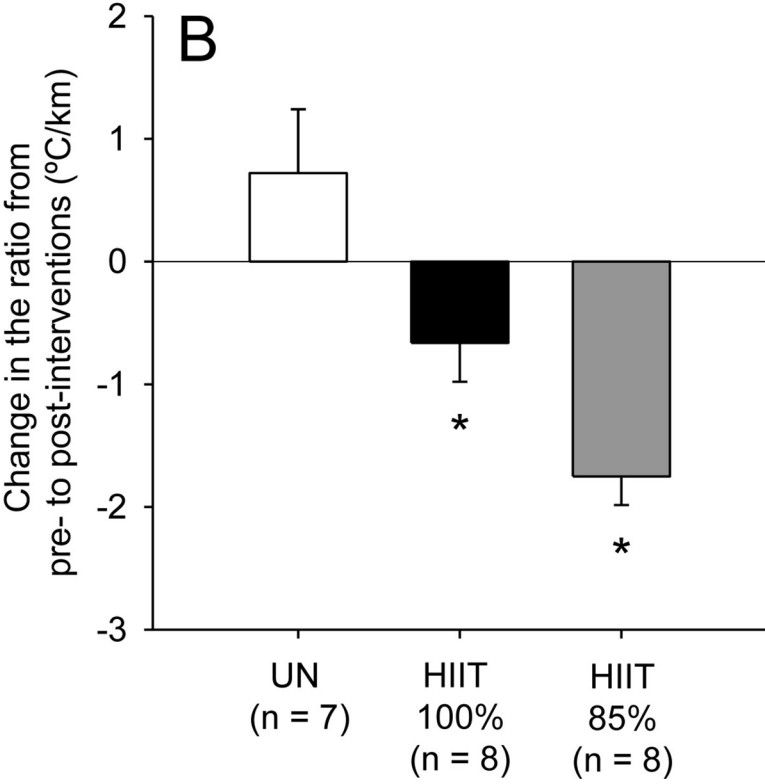

**Fig 8.** Ratio between the change in colonic temperature ($T_{COL}$) and the distance travelled in rats of the three experimental groups during the incremental exercises at pre- and post-interventions (panel A). Change in the ratio from pre- to post-interventions (panel B). The values are expressed as the means ± SEM. HIIT$_{100\%}$: high-intensity interval training at 100% of maximal speed; HIIT$_{85\%}$: high-intensity interval training at 85% of maximal speed; UN:

untrained, passive heat exposure. & denotes a significant difference from the first incremental exercise within the same group ($p < 0.05$); * denotes a significant difference from the UN group ($p < 0.05$).

approximately 40.5°C [8]. The improved thermoregulatory efficiency in our rats has most likely resulted from adaptations in cutaneous heat loss and/or in metabolic heat production.

In rats exposed to a warm environment, approximately 25–40% of whole body heat loss occurs through increases in cutaneous blood flow [37], which is dependent on an elaborate system of arteriovenous anastomoses in the tail skin [37,38]. However, changes in dry heat loss through the tail apparently did not underlie the improved thermoregulatory efficiency after the two HIIT protocols in our study. These findings confirm the observations that rats with high intrinsic aerobic capacity have greater thermoregulatory efficiency, but unchanged tail heat loss, relative to rats with standard and low intrinsic capacities [31]. In contrast, our findings disagree with the previous observations made by Santiago et al. [36], who observed an increased tail heat loss during exercise following an 8-week aerobic training consisting of constant-speed sessions. In the latter study [36], all training and testing were performed in a temperate environment, and training was initiated when rats were very young (i.e., 4 weeks old); these differences in methods may explain the different results yielded by the two studies. Particularly, dry heat exchange through the tail is dependent on cutaneous vasodilation that favours convection, which is more relevant in temperate than in hot environments due to a greater thermal gradient between the skin and surroundings under the former conditions.

A recent study highlighted that tail-skin vasodilation does not fully explain heat loss in running rats [39]. In this study, the rats subjected to tail artery denervation presented greatly impaired tail heat loss, but a "normal" exercise-induced increase in core temperature. Thus, differences in alternative pathways for dissipating heat other than the tail-skin vasodilation (e.g., ear vasodilation and water evaporation through respiration) could explain the improved thermoregulatory efficiency in trained rats. Ear vasodilation may be an alternative functional pathway for dissipating heat during exercise, although its effectiveness has never been investigated under such conditions. Regarding evaporative heat loss, the evaporation of water from the respiratory tract is dependent upon exercise intensity [18] and, therefore, may also contribute to thermoregulation in exercising rats. A facilitated evaporative heat loss may be particularly relevant for animals that will exercise in the heat, as evaporation is the only heat exchange pathway that is effective against a thermal gradient. Saliva-spreading behaviour is an important adjunct thermolytic mechanism, particularly in hot environments [40]. It is of note that the importance of this thermolytic pathway is minor during treadmill running because the rats are unable to spread saliva over their fur [41]. Taken together, the above-mentioned information indicates that we cannot rule out that trained rats have greater evaporative heat loss from the respiratory tract than UN rats, which might explain their differences in thermoregulatory efficiency.

We also suggest that mechanical efficiency is greater and, consequently, metabolic heat production is lower in trained rats. This suggestion is supported by a previous report showing that three different aerobic training protocols, consisting of submaximal, constant-speed running, improved mechanical efficiency in rats [21]. Aerobic training-induced changes in mechanical efficiency may result from central and peripheral adaptations. Central adaptations include, among others, altered neurotransmission in the caudate-putamen [42], an area involved in motor control and motivation to exercise; whereas peripheral adaptations include improved technique and the transfer of elastic energy during stretch-shortening cycles [43], as well an increased mitochondrial content, which results in augmented skeletal muscle respiratory capacity [44].

In conclusion, the two HIIT protocols used in the present study induce greater thermoregulatory adaptations and better aerobic performance than passive heat exposure. At least during the period investigated herein, the adaptations induced by heat acclimation (i.e., a better thermoregulatory efficiency and an improved performance during incremental exercise) are likely similar between the two HIIT protocols, even with differences in the number, intensity and duration of training sessions. Interestingly, the greater thermoregulatory efficiency in rats subjected to HIIT protocols occurs in the absence of an improved tail heat loss. More studies in rats are warranted to advance the knowledge about thermoregulatory adaptations resulting from different HIIT protocols, as well as the brain and molecular mechanisms underlying the adaptations induced by aerobic interval training and/or heat acclimation.

## Supporting information

**S1 Fig.** Panel A—Workload (J) during the training sessions. Panel B—Workload (J) during all training sessions.
(XLSX)

**S2 Fig.** Panel A—Colonic temperature (˚C) during the first heat acclimation session. Panel B—Tail-skin temperature (˚C) during the first heat acclimation session. Panel C—Colonic temperature (˚C) during the last heat acclimation session. Panel D—Tail-skin temperature (˚C) during the last heat acclimation session.
(XLSX)

**S3 Fig.** Panel A—Colonic temperature (˚C) at the beginning of the acclimation sessions. Panel B—Colonic temperature (˚C) at the end of the acclimation sessions. Panel C—Change in colonic temperature (˚C) during the acclimation sessions. Panel D—Tail-skin temperature (˚C) at the end of the acclimation sessions.
(XLSX)

**S4 Fig.** Panel A—Ratio between the change in colonic temperature and the distance travelled (˚C/km) during the acclimation sessions. Panel B—Ratio between the change in colonic temperature and the distance travelled (˚C/km) during the acclimation sessions in the HIIT 100% group. Panel C—Ratio between the change in colonic temperature and the distance travelled (˚C/km) during the acclimation sessions in the HIIT 85% group.
(XLSX)

**S5 Fig.** Panel A—Workload (J) performed during the incremental exercises before and after the two-weeks interventions. Panel B—Change in the workload (J) performed between the incremental exercises before and after the two-week interventions.
(XLSX)

**S6 Fig.** Panel A—Colonic temperature (˚C) during the first incremental exercise. Panel B—Tail-skin temperature (˚C) during the first incremental exercise. Panel C—Colonic temperature (˚C) during the second incremental exercise. Panel D—Tail-skin temperature (˚C) during the second incremental exercise.
(XLSX)

**S7 Fig.** Panel A—Colonic temperature (˚C) at the beginning of the incremental exercises. Panel B—Colonic temperature (˚C) at the end of the incremental exercises. Panel C—Change in colonic temperature (˚C) during the incremental exercises. Panel D—Tail-skin temperature (˚C) at the end of the incremental exercises.
(XLSX)

**S8 Fig.** Panel A—Ratio between the change in colonic temperature and the distance travelled (˚C/km) during the incremental exercises before and after the two-week interventions. Panel B—Change in the ratio between the change in colonic temperature and the distance travelled (˚C/km) between the incremental exercises before and after the two-week interventions. (XLSX)

## Author Contributions

**Conceptualization:** Myla Aguiar Bittencourt, Samuel Penna Wanner, Luiz Guilherme Antonacci Guglielmo.

**Formal analysis:** Myla Aguiar Bittencourt, Samuel Penna Wanner, Ana Cançado Kunstetter, Nicolas Henrique Santos Barbosa, Pedro Victor Ribeiro Andrade, Tiago Turnes, Luiz Guilherme Antonacci Guglielmo.

**Funding acquisition:** Samuel Penna Wanner, Luiz Guilherme Antonacci Guglielmo.

**Investigation:** Myla Aguiar Bittencourt, Samuel Penna Wanner, Ana Cançado Kunstetter, Nicolas Henrique Santos Barbosa, Paula Carolina Leite Walker, Tiago Turnes, Luiz Guilherme Antonacci Guglielmo.

**Methodology:** Myla Aguiar Bittencourt, Samuel Penna Wanner, Ana Cançado Kunstetter, Nicolas Henrique Santos Barbosa, Pedro Victor Ribeiro Andrade, Luiz Guilherme Antonacci Guglielmo.

**Project administration:** Myla Aguiar Bittencourt, Samuel Penna Wanner, Luiz Guilherme Antonacci Guglielmo.

**Resources:** Samuel Penna Wanner, Luiz Guilherme Antonacci Guglielmo.

**Supervision:** Samuel Penna Wanner, Luiz Guilherme Antonacci Guglielmo.

**Visualization:** Myla Aguiar Bittencourt, Samuel Penna Wanner, Luiz Guilherme Antonacci Guglielmo.

**Writing – original draft:** Myla Aguiar Bittencourt, Samuel Penna Wanner.

**Writing – review & editing:** Myla Aguiar Bittencourt, Samuel Penna Wanner, Ana Cançado Kunstetter, Nicolas Henrique Santos Barbosa, Paula Carolina Leite Walker, Pedro Victor Ribeiro Andrade, Tiago Turnes, Luiz Guilherme Antonacci Guglielmo.

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
