## [Decision Letter · Decision Letter 0]

19 Sep 2019

PONE-D-19-18686

Comparative effects of two heat acclimation protocols consisting of high-intensity interval training in the heat on aerobic performance and thermoregulatory responses in exercising rats

PLOS ONE

Dear Dr. Wanner,

Thank you for submitting your manuscript to PLOS ONE. After careful consideration, we feel that it has merit but does not fully meet PLOS ONE’s publication criteria as it currently stands. Therefore, we invite you to submit a revised version of the manuscript that addresses the points raised during the review process.

Dear Authors, 

Please address the major and minor comments from the reviewers. Specifically, there is need for greater clarity regarding hypothesis #2 and some study design issues/limitations

Thank you

We would appreciate receiving your revised manuscript by Nov 03 2019 11:59PM. To enhance the reproducibility of your results, we recommend that if applicable you deposit your laboratory protocols in protocols.io, where a protocol can be assigned its own identifier (DOI) such that it can be cited independently in the future. For instructions see: http://journals.plos.org/plosone/s/submission-guidelines#loc-laboratory-protocols

We look forward to receiving your revised manuscript.

Kind regards,

Matthew Cooke

Academic Editor

PLOS ONE

Journal Requirements:

2. Please include a copy of Table 2 which you refer to in your text on page 10.

Additional Editor Comments (if provided):

Dear Authors,

We have received the reviwers' feedback and would like you to address their comments.

Reviewers' comments:

Reviewer's Responses to Questions

**Comments to the Author**

1. Is the manuscript technically sound, and do the data support the conclusions?

Reviewer #1: Partly

Reviewer #2: Partly

2. Has the statistical analysis been performed appropriately and rigorously? 

Reviewer #1: No

Reviewer #2: Yes

3. Have the authors made all data underlying the findings in their manuscript fully available?

Reviewer #1: Yes

Reviewer #2: Yes

4. Is the manuscript presented in an intelligible fashion and written in standard English?

Reviewer #1: Yes

Reviewer #2: Yes

5. Review Comments to the Author

Reviewer #1: This manuscript describes a study into the effects of two high intensity interval training modalities on performance and thermoregulatory responses in rats in the heat. The manuscript is clear and well-written, and merit has to be given to the control of the study. However, it is difficult to ascertain whether the similarities/ differences observed between the two HIIT protocols are due to the variations in duration or intensity of exercise.

Major comment.

Changes in workload from pre to post interventions, although not statistically significant, appear to have a large effect between HIIT 100% and HIIT 85% (Figure 5). In performance studies, effect sizes are important as they provide a magnitude of difference, as opposed to a simple non-significance, which does not provide an extent of difference. Within this context, HIIT 85% may be practically better for performance based solely on Figure 5B. However, it is difficult to ascertain whether this effect was due to a longer duration (higher workload) in HIIT 85%, as the workloads between conditions were not matched (Figure 1B). As such, the current study design and results do not help to clarify the second hypothesis.

Specific comments.

Line 531

Indeed, it is difficult to determine the degree of influence that intensity had on thermoregulatory responses in this study.

Line 612

The suggestion and the paragraph should be related to a specific result measured in the study.

Line 632

It is normally true that higher intensity interval bouts are more time efficient in achieving work done. In this study, despite a lesser exercise time in the HIIT 100% compared with HIIT 85%, HIIT 85% appears to have a large performance effect over HIIT 100%. Moreover, colonic and tail temperatures were not different. Therefore, it is difficult to confidently conclude that HIIT 100% was more efficient in inducing heat acclimation.

Reviewer #2: Minor Points

Abstract

line 29, "exercise at" should read "exercise to fatigue at"

line 40, when compared to UN rats. I think that this would be better described "when compared to pre-intervention test". Using UN both within the brackets and the subsequent statement is confusing.

Introduction

Line 105-108

The second hypothesis, regarding comparisons between HIIT protocols. I do not understand the logic of the hypothesis based on the higher intensity creating a greater thermoregulatory strain. I think that that the total workload over a session would better reflect the total endogenous heat generation and thus with the lower intensity protocol design in this study seemingly performing more work over a training session.

Procedures

lines 176-178, I am a little confused by the Smax formula which doesnt make sense to me. Smax = S + (t/180). S is in units of m.min-1 yet the (t/180) which is in a time unit is added to it. How does the addition of these two factors reconcile to result in a unit of velocity for Smax. I know it is referenced but I dont have access to that article to check the original source.

Results

Figure 4

This is important in representing thermal load. However, I believe the units of the ratio axis in parentheses is wrong. It should be per m, not per min.

Figure 6. again the units. per m? not per min?

Major

Discussion

Clarity required

I think the discussion may be complicated by temporal and workload (intensity and bout number) differences in the training design. However the incremental protocol offers the most controlled comparison of the heat acclimation with the different exercise training programs, Figure 5 demonstrated the 85% intensity training produces the greatest workload improvement (not significant) in the incremental test protocol and the greatest attenuation of heat relative to the distance travelled (ie workload). This indicated thermodynamic efficiency is improved post acclimation in this group, which appear to be contrary to the conclusion. Considering this is the most controlled performance measure between the groups. More needs to be made of this combination of results in the discussion. It may be that whilst intensity is effective, work volume may be a more effective factor in acclimation process. Furthermore, clarity of the sentence beginning line 630 needs to be revisited. In particular, what is meant by more efficient strategy? when the results were not entirely comparable in terms of thermal changes as demonstrated in Fig 8 B

6. PLOS authors have the option to publish the peer review history of their article (what does this mean?). If published, this will include your full peer review and any attached files.

Reviewer #1: No

Reviewer #2: No

---

## [Author Response · Author response to Decision Letter 0]

11 Dec 2019

Reviewer #1

This manuscript describes a study into the effects of two high intensity interval training modalities on performance and thermoregulatory responses in rats in the heat. The manuscript is clear and well-written, and merit has to be given to the control of the study. However, it is difficult to ascertain whether the similarities/ differences observed between the two HIIT protocols are due to the variations in duration or intensity of exercise.

We thank the first reviewer for the constructive comments on our manuscript. Please see below our responses to these comments.

Major comment.

1- Changes in workload from pre to post interventions, although not statistically significant, appear to have a large effect between HIIT 100% and HIIT 85% (Figure 5). In performance studies, effect sizes are important as they provide a magnitude of difference, as opposed to a simple non-significance, which does not provide an extent of difference. Within this context, HIIT 85% may be practically better for performance based solely on Figure 5B. However, it is difficult to ascertain whether this effect was due to a longer duration (higher workload) in HIIT 85%, as the workloads between conditions were not matched (Figure 1B). As such, the current study design and results do not help to clarify the second hypothesis.

Thank you again for the very helpful comments. To address this major comment, we have made several changes to the manuscript, which are listed as follows: 

1- we have calculated effect sizes for most of the comparisons made between the two HIIT protocols (please see how the calculation was made in pages 14 and 15, lines 298 to 304). More specifically, effect size comparisons revealed that: 1- the training-induced increase in workload was greater in the HIIT85% than in the HIIT100% group (i.e., a moderate effect size, figure 5B); 2- the training-induced increase in thermoregulatory efficiency was also greater in the HIIT85% than in the HIIT100% group (i.e., a large effect size, figure 8B). This information was included in the revised manuscript: page 22, lines 450 to 453, and page 24, lines 510 to 513.

2- we have clarified our second hypothesis (page 6, lines 116 to 124). The revised hypothesis states that the improvement in aerobic performance was expected to be more evident after the HIIT100%, whereas thermoregulatory adaptations were expected to be more evident after the HIIT85%. 

3- we have amended the conclusion (page 33, lines 704 to 719), which is now formulated based on the results generated by our experiments.

4- we have acknowledged in the revised discussion that future experiments should compare the adaptations induced by HIIT protocols consisting of training sessions at different intensities but with the same total workload (pages 28 and 29, lines 597 to 600).

Specific comments.

2- Line 531. Indeed, it is difficult to determine the degree of influence that intensity had on thermoregulatory responses in this study.

We agree with the comment made by the reviewer, because our method does not allow us to isolate the influence of intensity on the adaptations induced by heat acclimation. To avoid erroneous interpretation of our data, we have thoroughly revised the discussion section. For example, the revised manuscript highlights a possible role of exercise / heat exposure duration in the induction of heat acclimation (page 28, lines 591 to 598). Moreover, as stated earlier, we have acknowledged in the revised discussion that future experiments should compare the adaptations induced by HIIT protocols consisting of training sessions at different intensities but with the same total workload (pages 28 and 29, lines 597 to 600).

3- Line 612. The suggestion and the paragraph should be related to a specific result measured in the study.

The information written in the penultimate paragraph of the discussion does not relate to any specific result generated by the present study. However, we believe that this paragraph is important, because it helps to address a question mentioned earlier in the discussion: “The improved thermoregulatory efficiency in our rats has most likely resulted from adaptations in cutaneous heat loss and/or in metabolic heat production”. In addition, this paragraph indicates to readers a possible experiment that should be performed in the future (i.e., the evaluation of running economy in rats subjected to HIIT protocols in the heat). Therefore, we kindly ask the reviewer to allow us to maintain this paragraph in the revised manuscript.

4- Line 632. It is normally true that higher intensity interval bouts are more time efficient in achieving work done. In this study, despite a lesser exercise time in the HIIT 100% compared with HIIT 85%, HIIT 85% appears to have a large performance effect over HIIT 100%. Moreover, colonic and tail temperatures were not different. Therefore, it is difficult to confidently conclude that HIIT 100% was more efficient in inducing heat acclimation.

We agree with the important comment made by the reviewer. The statement that “that HIIT 100% was more efficient in inducing heat acclimation” was deleted to address the issues raised by both reviewers (please see the revised conclusion in page 33, lines 704 to 719). In addition, we have made a great effort to improve the discussion of our findings. The revised manuscript no longer states that a lower thermoregulatory efficiency means a greater thermoregulatory strain. Instead, we have tried to discuss our thermoregulation data (colonic and tail-skin temperatures and thermoregulatory efficiency) in an integrated manner. 

Reviewer #2: 

Major Points

1. Discussion: Clarity required.

I think the discussion may be complicated by temporal and workload (intensity and bout number) differences in the training design. However the incremental protocol offers the most controlled comparison of the heat acclimation with the different exercise training programs, Figure 5 demonstrated the 85% intensity training produces the greatest workload improvement (not significant) in the incremental test protocol and the greatest attenuation of heat relative to the distance travelled (ie workload). This indicated thermodynamic efficiency is improved post acclimation in this group, which appear to be contrary to the conclusion. 

We thank the second reviewer for the constructive comments on our manuscript. Please see below our responses to these comments.

Regarding the data presented in figure 5, despite the lack of statistical differences between groups subjected to HIIT protocols, the change in workload induced by training was 51% greater in the HIIT85% than in the HIIT100% group. The greater value in the HIIT85% group can be classified as a moderate effect size (ES = 0.74). This information was inserted in the results section of the revised manuscript (page 22, lines 450 to 453). Please note that we have inserted effect size assessment (i.e., Cohen’s d) in the revised manuscript, as suggested by the first reviewer. 

Regarding the data presented in figure 8, we noticed a mistake in the figure that was originally submitted. The post hoc test (i.e., Tukey’s test) could not reveal a significant difference between the HIIT85% and HIIT100% groups (p = 0.100). We are sorry for this mistake, which was corrected in the revised version of figure 8. 

Therefore, despite the lack of statistical differences between groups subjected to HIIT protocols, the change in the ratio induced by training was 165% greater in the HIIT85% than in the HIIT100% group (Figure 8B); the greater value in the HIIT85% group can be classified as a large effect size (ES = 1.39). This information was inserted in the results section of the revised manuscript (page 24, lines 510 to 513). 

We agree that the result mentioned in the paragraph above is contrary to our original conclusion. Therefore, we have amended the conclusion to properly address the present findings (page 33, lines 704 to 719).

2. Considering this is the most controlled performance measure between the groups. More needs to be made of this combination of results in the discussion. It may be that whilst intensity is effective, work volume may be a more effective factor in acclimation process. 

We have made a great effort to improve the discussion of our findings. Of note, the revised manuscript no longer states that a lower thermoregulatory efficiency means a greater thermoregulatory strain. Instead, we have tried to discuss our thermoregulation data (colonic and tail-skin temperatures and thermoregulatory efficiency) in an integrated manner. For example, we have changed information in the following sentences / paragraphs: page 26, lines 530 to 540; page 27, lines 559 to 565; pages 28 and 29, lines 576 to 600.

3. Furthermore, clarity of the sentence beginning line 630 needs to be revisited. In particular, what is meant by more efficient strategy? when the results were not entirely comparable in terms of thermal changes as demonstrated in Fig 8 B

The statement that “that HIIT 100% was more efficient in inducing heat acclimation” was deleted to address the issues raised by both reviewers. We agree that information in this sentence was contradictory to data presented in figures 5 and 8. The revised conclusion is now formulated based on the results generated by our experiments (page 33, lines 704 to 719).

Minor Points

1. Abstract. Line 29, "exercise at" should read "exercise to fatigue at"

The sentence was changed as follows (page 2, line 29): “Twenty-three adult male Wistar rats were initially subjected to an incremental-speed exercise at 32�C until they were fatigued and then …”.

2. Abstract. Line 40, when compared to UN rats. I think that this would be better described "when compared to pre-intervention test". Using UN both within the brackets and the subsequent statement is confusing.

The abstract was revised to present the workload and ratio values calculated from the second incremental-speed exercise that was performed after the interventions. Therefore, the sentence was changed as follows (page 2, lines 36 to 42): “After the intervention period, rats subjected to both HIIT protocols attained greater workloads (HIIT100%: 313.7 ± 21.9 J vs. HIIT85%: 318.1 ± 32.6 J vs. UN: 250.8 ± 32.4 J; p < 0.05) and presented a lower ratio between the change in TCOL and the distance travelled (HIIT100%: 4.95 ± 0.42 �C/min vs. HIIT85%: 4.33 ± 0.59 �C/min vs. UN: 6.14 ± 1.03 �C/min; p < 0.001) when compared to UN rats.” We hope that this change has improved clarity of the abstract. Moreover, please note that the abstract was further edited to reduce its size, thereby not exceeding the 300-word limit.

3. Introduction. Line 105-108

The second hypothesis, regarding comparisons between HIIT protocols. I do not understand the logic of the hypothesis based on the higher intensity creating a greater thermoregulatory strain. I think that that the total workload over a session would better reflect the total endogenous heat generation and thus with the lower intensity protocol design in this study seemingly performing more work over a training session.

Thank you for the very good comment. The text in the introduction was changed to explain the effects of exercise intensity and duration on thermoregulatory responses, particularly under conditions of uncompensable heat stress (pages 5 and 6, lines 95 to 102). Moreover, the second hypothesis was revised: it now states that the improvement in aerobic performance was expected to be more evident after the HIIT100%, whereas thermoregulatory adaptations were expected to be more evident after the HIIT85% (page 6, lines 116 to 124). 

4. Procedures. lines 176-178.

I am a little confused by the Smax formula which doesnt make sense to me. Smax = S + (t/180). S is in units of m.min-1 yet the (t/180) which is in a time unit is added to it. How does the addition of these two factors reconcile to result in a unit of velocity for Smax. I know it is referenced but I dont have access to that article to check the original source.

Thank for the suggestion. The equation was amended to improve its clarity. The amended equation is presented as follows (page 12, lines 232 to 235): SMAX = S + (t1 / t2), where S = speed in the last completed stage in m/min; and t1 = time spent in the incomplete stage in seconds; and t2 = duration of each stage, which corresponds to 180 seconds. 

Therefore, t1 and t2 are both expressed in seconds. Thus, the second factor of the equation corresponds to seconds divided by seconds and, therefore, has no unit. 

5. Results. Figure 4.

This is important in representing thermal load. However, I believe the units of the ratio axis in parentheses is wrong. It should be per m, not per min.

The reviewer is correct. We are sorry for this mistake. The units inside the parentheses were corrected in figure 4.

6. Results. Figure 6. again the units. per m? not per min?

If this comment is related to figure 8, the reviewer is correct again. Again, we are sorry for this mistake. The units inside the parentheses were corrected in figure 8.

---

## [Decision Letter · Decision Letter 1]

17 Jan 2020

PONE-D-19-18686R1

Comparative effects of two heat acclimation protocols consisting of high-intensity interval training in the heat on aerobic performance and thermoregulatory responses in exercising rats

PLOS ONE

Dear Dr. Wanner,

Thank you for submitting your manuscript to PLOS ONE. After careful consideration, we feel that it has merit but does not fully meet PLOS ONE’s publication criteria as it currently stands. Therefore, we invite you to submit a revised version of the manuscript that addresses the points raised during the review process.

Thank you for addressing the reviwers's comments. Given the error that occured with the reporting of some data, the reviewer would like this to be addressed in the discussion. Please refer to their comments

We would appreciate receiving your revised manuscript by Mar 02 2020 11:59PM. To enhance the reproducibility of your results, we recommend that if applicable you deposit your laboratory protocols in protocols.io, where a protocol can be assigned its own identifier (DOI) such that it can be cited independently in the future. For instructions see: http://journals.plos.org/plosone/s/submission-guidelines#loc-laboratory-protocols

We look forward to receiving your revised manuscript.

Kind regards,

Matthew Cooke

Academic Editor

PLOS ONE

Reviewers' comments:

Reviewer's Responses to Questions

**Comments to the Author**

1. If the authors have adequately addressed your comments raised in a previous round of review and you feel that this manuscript is now acceptable for publication, you may indicate that here to bypass the “Comments to the Author” section, enter your conflict of interest statement in the “Confidential to Editor” section, and submit your "Accept" recommendation.

Reviewer #1: All comments have been addressed

Reviewer #2: All comments have been addressed

2. Is the manuscript technically sound, and do the data support the conclusions?

Reviewer #1: Yes

Reviewer #2: Partly

3. Has the statistical analysis been performed appropriately and rigorously? 

Reviewer #1: Yes

Reviewer #2: Yes

4. Have the authors made all data underlying the findings in their manuscript fully available?

Reviewer #1: Yes

Reviewer #2: Yes

5. Is the manuscript presented in an intelligible fashion and written in standard English?

Reviewer #1: Yes

Reviewer #2: Yes

6. Review Comments to the Author

Reviewer #1: Thank you for thoroughly addressing comments suggested.

Please ensure data is uploaded and available.

Reviewer #2: I believe the revision has tightened the manuscript. However, considering the rectification of the error, I believe that now the importance of non HIIT protocols becomes more relevant. The endothermic load of exercise may be all that is needed for acclimation, constant workload of HIT protocols (any protocol). It may just be a question of exposure to elevated temperature for long enough period of time to stimulate adaptations. This is not an insignificant point. I think a section addressing this point in the discussion may be warranted as there is no difference between these protocols.

7. PLOS authors have the option to publish the peer review history of their article (what does this mean?). If published, this will include your full peer review and any attached files.

Reviewer #1: Yes: Sam Wu

Reviewer #2: Yes: Christos George Stathis

---

## [Author Response · Author response to Decision Letter 1]

27 Jan 2020

Dear editor and reviewers,

Thank you for the review of the manuscript entitled “Comparative effects of two heat acclimation protocols consisting of high-intensity interval training in the heat on aerobic performance and thermoregulatory responses in exercising rats”. We have carefully addressed the comment from the second reviewer and then modified the manuscript accordingly; changes made to the manuscript are highlighted in the revised version with track changes. We believe that the revised version of our manuscript has been improved and hope that it will be suitable for publication.

P.S.: the page and line numbers indicated in the answer below correspond to the revised manuscript with track changes.

Reviewer #1

Thank you for thoroughly addressing comments suggested. Please ensure data is uploaded and available.

We thank the first reviewer for the positive comment. Moreover, we confirm that all data were uploaded in Microsoft Excel spreadsheet format and are available for readers.

Reviewer #2: 

I believe the revision has tightened the manuscript. However, considering the rectification of the error, I believe that now the importance of non HIIT protocols becomes more relevant. The endothermic load of exercise may be all that is needed for acclimation, constant workload of HIT protocols (any protocol). It may just be a question of exposure to elevated temperature for long enough period of time to stimulate adaptations. This is not an insignificant point. I think a section addressing this point in the discussion may be warranted as there is no difference between these protocols.

We thank the second reviewer for the constructive comment. A paragraph was inserted in the discussion (page 29, lines 595-607) to address his/her comment. This paragraphs highlights that: a- the endothermic heat production during exercise and the resulting body heat storage may drive acclimation in rats; b- thermoregulatory adaptations likely result from different combinations among environmental heat stress and exercise intensity and duration; c- depending on how training sessions are planned, non HIIT protocols (e.g., moderate-intensity continuous training) can be effective in inducing heat acclimation in rats.

---

## [Decision Letter · Decision Letter 2]

5 Feb 2020

Comparative effects of two heat acclimation protocols consisting of high-intensity interval training in the heat on aerobic performance and thermoregulatory responses in exercising rats

PONE-D-19-18686R2

Dear Dr. Wanner,

We are pleased to inform you that your manuscript has been judged scientifically suitable for publication and will be formally accepted for publication once it complies with all outstanding technical requirements.

With kind regards,

Matthew Cooke

Academic Editor

PLOS ONE

Additional Editor Comments (optional):

Reviewers' comments:

Reviewer's Responses to Questions

**Comments to the Author**

1. If the authors have adequately addressed your comments raised in a previous round of review and you feel that this manuscript is now acceptable for publication, you may indicate that here to bypass the “Comments to the Author” section, enter your conflict of interest statement in the “Confidential to Editor” section, and submit your "Accept" recommendation.

Reviewer #2: All comments have been addressed

2. Is the manuscript technically sound, and do the data support the conclusions?

Reviewer #2: Yes

3. Has the statistical analysis been performed appropriately and rigorously? 

Reviewer #2: I Don't Know

4. Have the authors made all data underlying the findings in their manuscript fully available?

Reviewer #2: Yes

5. Is the manuscript presented in an intelligible fashion and written in standard English?

Reviewer #2: Yes

6. Review Comments to the Author

Reviewer #2: (No Response)

7. PLOS authors have the option to publish the peer review history of their article (what does this mean?). If published, this will include your full peer review and any attached files.

Reviewer #2: Yes: Christos G. Stathis

---

## [Editor Report · Acceptance letter]

10 Feb 2020

PONE-D-19-18686R2 

Comparative effects of two heat acclimation protocols consisting of high-intensity interval training in the heat on aerobic performance and thermoregulatory responses in exercising rats 

Dear Dr. Wanner:

I am pleased to inform you that your manuscript has been deemed suitable for publication in PLOS ONE. Congratulations! Your manuscript is now with our production department. 

With kind regards,

on behalf of

Dr. Matthew Cooke 

Academic Editor

PLOS ONE